# Rapid P-TEFb-dependent transcriptional reorganization underpins the glioma adaptive response to radiotherapy

Faye M. Walker[1,10], Lays Martin Sobral[1,10], Etienne Danis[2,3], Bridget Sanford[1], Sahiti Donthula[1], Ilango Balakrishnan[1], Dong Wang[1], Angela Pierce[1], Sana D. Karam ®[4], Soudabeh Kargar[3], Natalie J. Serkova[5], Nicholas K. Foreman[1,6,7], Sujatha Venkataraman[1], Robin Dowell ®[8,9], Rajeev Vibhakar[1,6,7] & Nathan A. Dahl ®[1,6] ✉

Dynamic regulation of gene expression is fundamental for cellular adaptation to exogenous stressors. P-TEFb-mediated pause-release of RNA polymerase II (Pol II) is a conserved regulatory mechanism for synchronous transcriptional induction in response to heat shock, but this pro-survival role has not been examined in the applied context of cancer therapy. Using model systems of pediatric high-grade glioma, we show that rapid genome-wide reorganization of active chromatin facilitates P-TEFb-mediated nascent transcriptional induction within hours of exposure to therapeutic ionizing radiation. Concurrent inhibition of P-TEFb disrupts this chromatin reorganization and blunts transcriptional induction, abrogating key adaptive programs such as DNA damage repair and cell cycle regulation. This combination demonstrates a potent, synergistic therapeutic potential agnostic of glioma subtype, leading to a marked induction of tumor cell apoptosis and prolongation of xenograft survival. These studies reveal a central role for P-TEFb underpinning the early adaptive response to radiotherapy, opening avenues for combinatorial treatment in these lethal malignancies.

Adaptation and fine-tuning of gene expression are required to ensure appropriate cellular responses to developmental signals or environmental stressors. Neoplastic cells incorporate numerous signaling inputs in order to calibrate their net transcriptional output, including both cell state- and oncogene-specific intrinsic programs as well as extrinsic pressures from the tumor microenvironment, the immune system, and anti-cancer therapies. One of the most ubiquitously employed modalities in cancer is ionizing radiation (IR), in which genomic integrity is disrupted through the generation of DNA double-strand breaks[1]. The adaptive response to this cellular injury involves a coordinated interplay between DNA repair mechanisms and cell cycle regulation, collectively termed the DNA damage response (DDR)[1–3]. In cancer, effective DDR facilitates recovery from therapeutic IR and ultimately disease recurrence after treatment[4–7]. This signaling network has been exhaustively well characterized, transduced through key regulatory proteins such ATM, ATR, and p53[2]. Much of this

[1]Morgan Adams Foundation Pediatric Brain Tumor Research Program, Department of Pediatrics, University of Colorado School of Medicine, Aurora, CO, USA. [2]Department of Biomedical Informatics, University of Colorado School of Medicine, Aurora, CO, USA. [3]University of Colorado Cancer Center, University of Colorado School of Medicine, Aurora, CO, USA. [4]Department of Radiation Oncology, University of Colorado School of Medicine, Aurora, CO, USA. [5]Department of Radiology, University of Colorado School of Medicine, Aurora, CO, USA. [6]Center for Cancer and Blood Disorders, Children's Hospital Colorado, Aurora, CO, USA. [7]Department of Neurosurgery, University of Colorado School of Medicine, Aurora, CO, USA. [8]Department of Molecular, Cellular, and Developmental Biology, University of Colorado, Boulder, CO, USA. [9]BioFrontiers Institute, University of Colorado, Boulder, CO, USA. [10]These authors contributed equally: Faye M. Walker, Lays Martin Sobral. ✉e-mail: nathan.dahl@childrenscolorado.org

response occurs at a proteomic level, but robust transcriptomic changes are required to both initiate and sustain this corrective program[8–10]. And underpinning the control of gene expression is an array of epigenetic inputs such as chromatin state and transcriptional cofactors, which collectively regulate the recruitment, initiation, and productive elongation of RNA polymerase II (Pol II).

Focal chromatin state is the landscape over which transcription proceeds, including the physical accessibility of chromatin-binding factors to DNA as well as the net epigenetic signal conveyed by histone tail post-translational modifications (PTMs)[11,12]. Many epigenetically active loci are relatively stable within a given cell, reflecting lineage commitment and developmental cell identity[13,14]. But others are highly dynamic, reorganizing active chromatin to facilitate the transcriptional changes necessary to respond to a given stimulus[15]. For example, fibroblasts have been shown to dramatically redistribute chromatin accessibility and H3K27ac over time following UV exposure, remodeling enhancers and super-enhancers in response to environmental pressure[16]. Similar responses have been described across cell types and exogenous stimuli, highlighting the fundamental role of epigenetic signaling in driving adaptive transcriptional programs[8,9,17,18].

The functional endpoint of this epigenetic reorganization is productive transcriptional elongation by Pol II at the appropriate genomic loci. Processive transcription by Pol II is controlled through dynamic phosphorylation of the highly conserved C-terminal domain (CTD)[19–21]. Following Pol II recruitment, initiation is governed by the pre-initiation complex, in which the CDK7-containing TFIIH phosphorylates the CTD at the Ser5 position[22]. After transcribing 20-80 bases downstream, Pol II then becomes paused in promoter-proximal regions at many genomic loci. Phosphorylation of the CTD Ser2 by the CDK9/CyclinT1 pair, collectively termed positive transcription elongation factor b (P-TEFb), is then required for pause release[21,23–25]. This paused state can then represent a rate-limiting regulatory intermediate. P-TEFb is largely constitutively expressed[26], with its primary mode of regulation being sequestration in and controlled release from an inactivated pool where it is bound by the inhibitory 7SK small nuclear riboprotein (7SK snRNP) and the associated RNA-binding proteins MePCE, LARP7, and HEXIM1[27,28]. Activation of P-TEFb via release from 7SK snRNP to then coordinate Pol II pause release is well defined in models of exogenous stressors such as heat shock[29,30], genotoxic stress[27], and developmental signaling[31,32], while misregulation of this checkpoint has been implicated in cancer and other human disease[33–37]. Much has been learned about the mechanistic biology contributing to transcriptional elongation checkpoint control, but the therapeutic potential for its judicious disruption remains largely unrealized.

Pediatric high-grade gliomas (HGGs), including diffuse intrinsic pontine gliomas (DIPGs) and other histone-mutant diffuse midline gliomas (DMGs), are aggressive malignancies of childhood for which radiation therapy remains the only standard of care, and HGG relapse after radiotherapy represents a uniformly fatal event. Here, we examine the dynamics of chromatin reorganization and P-TEFb-mediated transcriptional induction in response to radiotherapy in pediatric HGG, and we investigate whether pharmacologic disruption of this adaptive reprogramming can be employed for therapeutic effect.

## Results

### Glioma cells rapidly reorganize active chromatin following exposure to IR

The physical accessibility of DNA determines permissible interactions by chromatin-binding factors to collectively regulate gene expression. The landscape of accessibility is rapidly and dynamically regulated in response to both extrinsic stimuli and developmental cues, and it can be measured by the susceptibility of DNA to cleavage using techniques such as the assay for transposase-accessible chromatin using sequencing (ATAC-seq) (reviewed by Klemm et al.[11]). We first sought to characterize the early reorganization of accessibility that we

hypothesized would occur immediately following radiotherapy. In contrast to adult diffuse gliomas, pediatric gliomas are frequently defined by recurrent mutations to histone *H3F3A* or *HIST1H3B* genes[38,39]. *TP53* mutations may variably occur as secondary events[40] but convey well-characterized alterations to the DNA damage response[8,10,41–43]. Therefore, we selected the well-characterized SU-DIPG-IV (*HIST1H3B*[mut], *ACVR1*[G328V], *TP53*[WT]) culture model for this study. We first performed ATAC-seq both at resting state and four hours following a single exposure to 6 Gy IR. This revealed a rapid shift biased directionally towards a more compacted chromatin state, with 3,237 peaks differentially lost and only 428 peaks differentially gained (Fig. 1a, b). These accessibility losses occurred primarily at annotated promoters (79%, $n = 2558$), with only 21% ($n = 679$) occurring at enhancers or other non-promoter transcription start sites (TSS). Ontology analysis of these loci showed functional enrichment in networks governing transcriptional processing and chromatin remodeling (e.g. *MED* subunits, *RBPJ*, *CHD4*, *EZH2*), DDR programs such as DNA repair and cell cycle regulation (*DUSP1*, *PPM1D*, *WEE1*), and cell structure morphogenesis (*NRCAM*, *STMN1*, *NPTN*) (Fig. 1c and Supplementary Data 1a). These signatures suggest a selectivity for this chromatin compaction at loci functionally correlated with early adaptive cellular reorganization and DDR activation. Ontology analysis of differentially gained peaks yielded comparatively weak enrichment across non-specific terms (Supplementary Data 1b). TRRUST analysis of transcriptional regulatory networks[44,45] within loci of differential accessibility inferred TP53 as the strongest upstream regulator affecting these changes, followed by HIF1A and E2F1 (Fig. 1d).

In order to better characterize active chromatin features which might predict changes in transcriptional output, we then performed ChIP-seq for H3K27ac, an epigenetic marker of transcriptionally active chromatin, in the same experimental conditions. In contrast to the largely unidirectional shift in accessibility, early H3K27ac redistribution was balanced, with 3,536 peaks differentially gained and 3,195 peaks differentially lost (Fig. 1e). Peaks gained were almost exclusively at annotated promoters (94%, $n = 3328$), whereas losses were distributed across genomic elements (TSS 35%, $n = 1132$ vs eTSS 65%, $n = 2063$). When examined in relationship to the prior accessibility changes, the few regions of accessibility gains paradoxically lost acetylation, consistent with the nonspecific or secondary nature of this small subset of genomic elements. In contrast, the predominant cluster of promoters which underwent physical compaction after IR maintained comparatively stable levels of net acetylation, instead exhibiting a balanced redistribution of H3K27ac gains and losses across these regions (Fig. 1f). Ontology analysis of differential H3K27ac peaks showed similar functional enrichments across networks mediating DDR programs, cell cycle and growth, and cell morphogenesis. Intriguingly, we observed a striking overlap in biological processes enriched in both H3K27ac gains and losses as opposed to distinct terms differentially up- or downregulated from these gene lists (Fig. 1g and Supplementary Data 2). Taken together, these data reveal a rapid shift towards chromatin compaction within hours of exposure to therapeutic IR. This loss of accessibility is neither uniform nor random, but rather it is preferentially enriched in functional programs one would predict would be transcriptionally relevant in an early cellular DNA damage response. Within this shifting landscape of permissible physical interaction, we observe a broad redistribution of H3K27ac at gene promoters, creating a framework of active chromatin over which differential recruitment and activation of transcriptional machinery might now occur.

### Redistribution of H3K27ac occupancy correlates with early differential transcript expression

Prior work has demonstrated that DNA damaging events transiently decrease elongation rates, phenocopying slow Pol II mutants[9,46]. This slowdown has functional implications, as Pol II elongation kinetics can

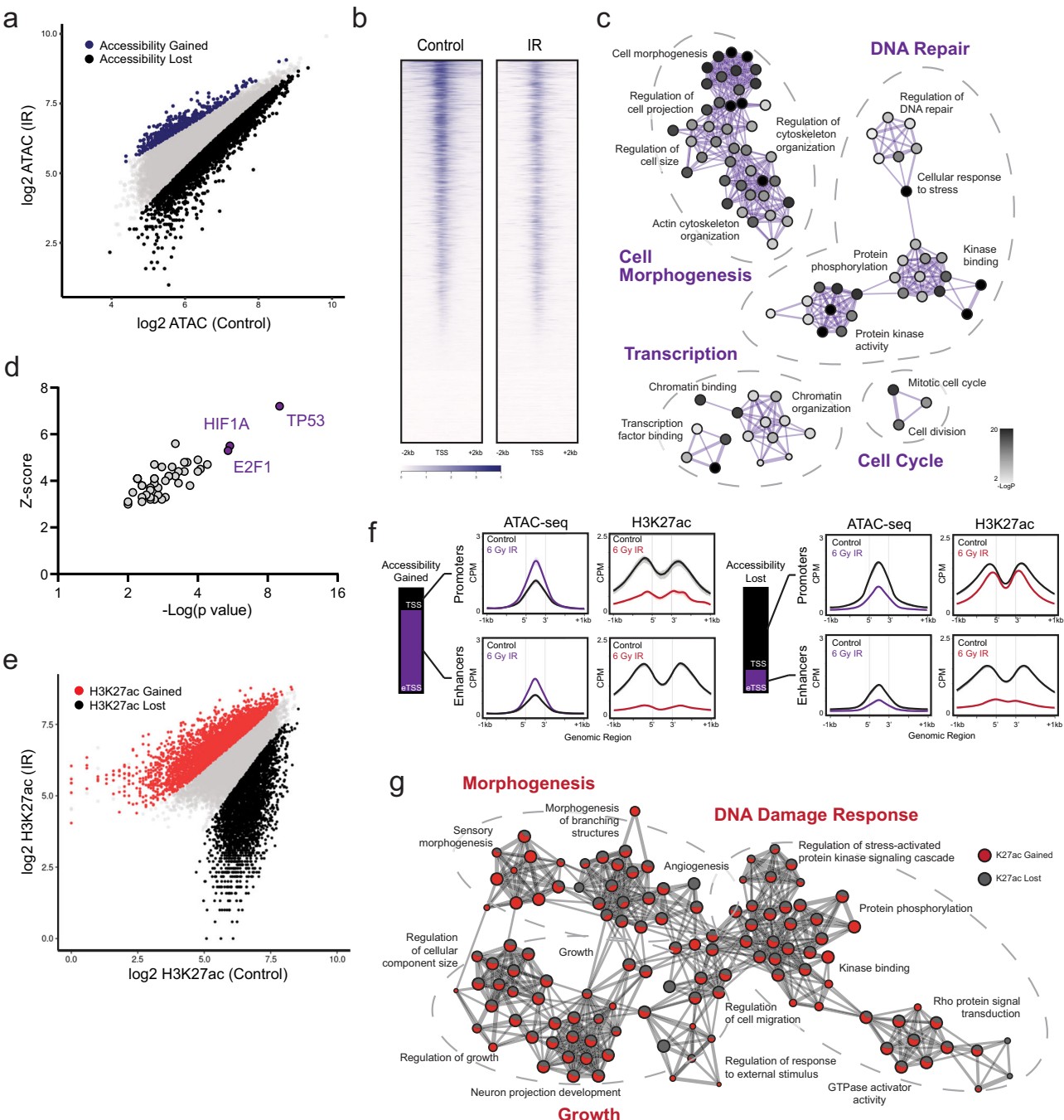

**Fig. 1 | Glioma cells rapidly reorganize active chromatin following exposure to IR. a** Scatterplot of ATAC-seq peaks compared between 6 Gy IR-exposed cells and untreated controls ($n = 2$). Differentially gained loci of accessibility are indicated in purple, differentially lost accessibility in black. **b** Genome-wide heatmap of accessibility before and after IR exposure. **c** Gene ontology network constructed from loci of accessibility lost following IR. Each node denotes an enriched term, with color density reflecting -log(Pval). **d** TRRUST inference of transcription factor-target pairs from differential ATAC-seq loci, ranked by Fisher's exact test (-Log(p val)). **e** Scatterplot of H3K27ac ChIP-seq peaks compared between 6 Gy IR-exposed cells and untreated controls ($n = 2$). Differentially gained H3K27ac occupancy is indicated in red, differentially lost occupancy in black. **f** H3K27ac ChIP-seq metagene profiles clustered by ATAC-seq-defined chromatin features. Shaded bands reflect standard error. **g** Gene ontology network constructed from loci of differential H3K27ac occupancy following IR. Each node denotes an enriched term, with color ratio reflecting relative contribution from H3K27ac gain and lost lists. Source data are provided as a Source Data file.

alter mRNA co-transcriptional processing, alternative splicing, and alternative isoform expression[9,46,47]. Consistent with this, in situ fluorescent staining of nascent RNA synthesis showed a time-dependent decrease in net transcriptional output following a single IR exposure (Fig. 2a). While this slowing enables pro-survival functions such as

repair of Pol II-encountered DNA lesions and limiting mutagenesis, it is held in opposition with the need to rapidly activate transcriptional programs involved in early DDR[8,9].

H3K27 acetylation functions as a transcriptionally activating modification in large part by serving as a target for transcription

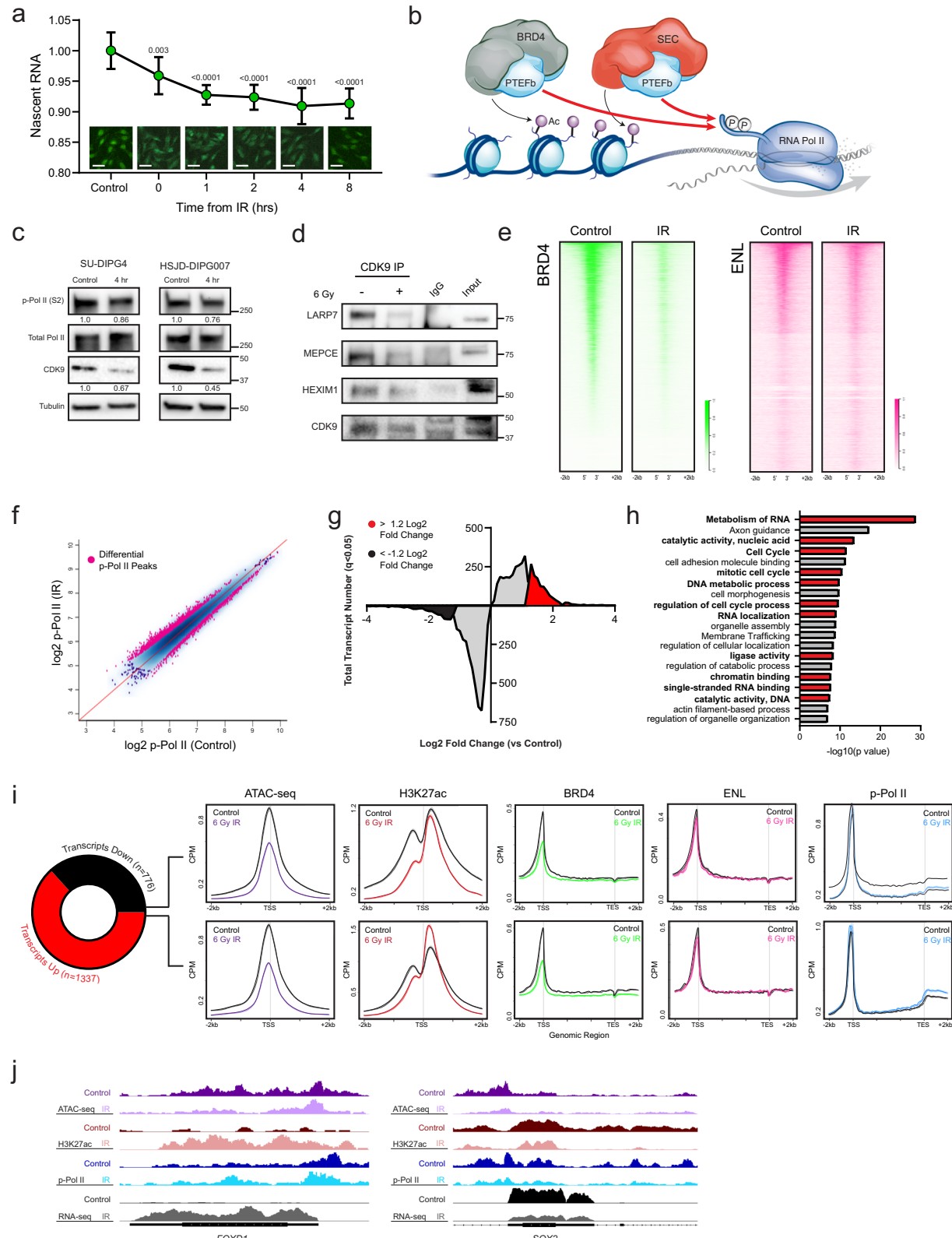

cofactor binding. This includes distinct P-TEFb-containing protein complexes such as the bromodomain and extraterminal domain family protein BRD4 and the super elongation complex (SEC)[48–53]. Once released from the inactivating 7SK snRNP pool[27,28] and recruited in a catalytically active complex, the CDK9 subunit of P-TEFb phosphorylates the CTD of Pol II at the serine 2 position, signaling for release into the gene body for productive elongation[19–21,54] (Fig. 2b). Given the

complex reordering of active chromatin we observed, we hypothesized that a similarly nuanced reorganization of transcriptional machinery might drive DDR programs within the context of overall transcriptional slowing.

We first examined immunoblots of CDK9 expression and the CDK9-catalyzed Pol II (Ser2) phosphorylation mark at the same early timepoint as the above chromatin studies. These showed a modest

**Fig. 2 | Redistributed H3K27ac occupancy correlates with early transcription from DDR programs. a** Click-IT fluorescent assay of relative nascent RNA abundance at indicated timepoints following 6 Gy IR. Comparisons reflect *p* value of two-tailed Student's t-test vs untreated control (bar = 50 μm), mean ± SD of *n* = 3 biologically independent replicates imaged 4 fields per replicate. **b**. Schematic representation of P-TEFb localization to H3K27ac-marked chromatin by active BRD4- or SEC-P-TEFb complexes to facilitate the phosphorylation of Pol II CTD (Ser2). **c** Immunoblot of p-Pol II (Ser2), total Pol II, and CDK9 measured 4 h after 6 Gy IR. Value below represents mean quantification of biological triplicates. **d** Immunoblot for 7SK snRNP complex members LARP7, MEPCE, and HEXIM1 following CDK9 co-immunoprecipitation before and 4 h after 6 Gy IR exposure. Data represent two independent experiments. **e** Genome-wide heatmap of BRD4 (left) and ENL (right) CUT&RUN occupancy before and after IR exposure (*n* = 2). **f** Scatterplot of p-Pol II (S2) CUT&RUN peaks compared between IR-exposed cells

and untreated controls (*n* = 3). Differentially bound peaks are indicated in pink. **g** Histogram of differentially expressed transcripts following IR. Transcripts with significant (Wilcoxon rank sum qval <0.05) but <1.2 LF change are indicated in grey. Transcripts with >± LFC are in red and black, respectively. **h** Functional ontology enrichment of transcripts ≥ 1.2 LFC in **e**. Unbiased top 20 terms identified by Metascape using a hypergeometric test and Benjamini-Hochberg *P* value correction algorithm are displayed, with terms involved in transcriptional processing or DDR in red. **i** Metagene plots of ATAC-seq, H3K27ac ChIP-seq, BRD4, ENL, and p-Pol II (S2) CUT&RUN changes at differentially expressed transcripts. **j** Illustrative loci at *FOXD1* and *SOX2* promoters demonstrate p-Pol II downstream egress and active transcription correlates with H3K27ac deposition irrespective of change in accessibility. Paired tracks reflect the same data scale. Source data are provided as a Source Data file.

decrease in total CDK9 expression and Pol II activation, respectively, concordant with net measures of transcriptional output (Fig. 2c). To examine P-TEFb regulation more specifically, we immunoprecipitated CDK9 before and after IR exposure to probe for association with LARP7, HEXIM1, and MEPCE, protein components of the 7SK snRNP. This demonstrated a decrease in 7SK snRNP member association, consistent with P-TEFb release (Fig. 2d). A degradation of total LARP7 was not observed (Supplementary Fig. 1), despite being reported in other model systems[55].

In order to then resolve occupancy profiles of CDK9 activity on chromatin, we next performed cleavage under targets and release using nuclease (CUT&RUN) for BRD4, ENL (the YEATS domain protein member of SEC)[56], and Ser2-phosphorylated Pol II. Occupancy of BRD4 on chromatin was significantly diminished following IR exposure (mean 52,213 peaks vs. 12,476, control vs. IR), while net ENL binding was comparatively preserved (mean 4144 peaks vs. 6134, control vs. IR) (Fig. 2e and Supplementary Fig. 2a, b). p-Pol II (Ser2) activation showed a striking redistribution across the genome within hours of IR exposure (1,923 reproducible, differential peaks *p* < 0.05, *n* = 3 replicates) (Fig. 2f and Supplementary Fig. 2c, d). The net change in p-Pol II chromatin occupancy was not significant using multiple peak calling parameters, suggesting that overall levels of Pol II activation are largely stable in this early time period (Supplementary Fig. 2e). To define the transcriptional output, we utilized transcriptome sequencing (RNA-seq) in the same conditions. This again demonstrated a greater number of unique transcripts differentially downregulated than up (4,775 down vs 3,677 up, qval <0.05, *n* = 3 replicates). In most cases, however, these downregulations were modest, falling below the 1.2 LFC cutoff commonly utilized for differential expression analyses. When examining only transcripts with differential LFC > ± 1.2, this pattern was reversed, with a greater number of transcripts strongly induced than repressed (1,337 up vs 776 down, qval <0.05) (Fig. 2g). Gene ontology enrichment analysis of these upregulated transcripts revealed marked activation of programs involved in transcriptional processing, cell cycle regulation, and DNA catalytic activity, consistent with a focal induction of critical DDR programs despite global transcriptional slowing (Fig. 2h and Supplementary Data 3).

We then intersected these RNA-seq data with the ATAC-seq, H3K27ac, BRD4, ENL, and p-Pol II datasets. Surprisingly, differential expression by RNA-seq showed no correlation with changes in accessibility, as both up- and downregulated transcripts showed similar degrees of chromatin compaction following IR exposure (Fig. 2i). This suggests that the primary function of this compaction is not in modulating gene expression, but it may instead reflect the protective role of a more heterochromatin state against subsequent DNA damage[57,58] for loci that must maintain active transcriptional output. In contrast, changes in H3K27ac deposition did exhibit a correlation with differential transcript level, with ENL binding and an increase in the P-TEFb-catalyzed p-Pol II pause-release and egress into downstream gene bodies observed at these loci (Fig. 2I, j). The discordance observed

between loss of BRD4 binding and increased transcription at these IR-induced genes may suggest a preference for SEC-P-TEFb in driving this response. This model of clear specificity while retaining some degree of redundancy amongst P-TEFb-containing complexes has been elegantly defined in models of heat shock[29], though more granular perturbation would be required to definitively delineate it here. Regardless, these data demonstrate a central role for H3K27ac redistribution in organizing P-TEFb-driven transcriptional output, including focal induction of critical DDR programs, within the broad chromatin compaction and transcriptional slowing observed following IR-induced genotoxic stress.

## P-TEFb inhibition disrupts IR-induced chromatin reorganization and abrogates transcriptional induction

Given the role of H3K27ac in recruiting active P-TEFb-containing complexes to drive Pol II pause-release[48–53], we hypothesized that selective inhibition of CDK9 should disrupt this adaptive response. To test this, we utilized AZD4573, a highly selective inhibitor of CDK9 with rapid target engagement, CDK9 enzymatic $IC_{50}$ of <0.003 μmol/L, and >25-fold cellular selectivity for CDK9 versus other CDKs[59]. Within our model system, we confirmed that AZD4573 treatment led to a dose-dependent depletion of both the CDK9-catalyzed Pol II Ser2 phosphorylation mark and nascent RNA synthesis (Fig. 3a, b), supporting its utility for this perturbation.

We then repeated the ATAC-seq, H3K27ac ChIP-seq, p-Pol II CUT&RUN, and RNA-seq experiments in the presence of AZD4573 co-treatment (*n* = 2–3, 40 nM dosed 2 h prior to irradiation) and examined these in comparison to our delineated IR-induced reorganization. Remarkably, the ATAC-defined chromatin compaction observed at both enhancers and promotors after radiation was almost completely abrogated in the presence of concurrent P-TEFb inhibition (Fig. 3c). The small subset of loci, which gained accessibility following radiation, were comparatively unaffected by AZD4573 treatment. Comparison of differential ATAC-seq peaks between IR- and AZD + IR-treated conditions identified 2960 differential loci, with ontology analysis yielding similar enrichment in programs related to stress response, transcriptional processing, and cell cycle regulation (Supplementary Data 1c). This suggests that the adaptive chromatin compaction observed following IR not only occurs independent of a transcriptional regulatory function, but that it is instead dependent on functional P-TEFb catalytic activity. This might reflect a direct effect of processive transcriptional activity on local chromatin architecture. Alternatively, prior work has shown that independent of its canonical role in modifying the Pol II CTD, P-TEFb indirectly modifies chromatin compaction via phosphorylation of the chromatin remodeling SWI/SNF complex member protein BRG1[60]. In that study, inhibition of CDK9 activity led to a loss of BRG1 phosphorylation and a subsequent chromatin relaxation in a BRG1-dependent manner[60]. To explore this as a mechanistic explanation for the observed accessibility changes, we performed immunoblot for phospho- (Ser1627/1631) and total BRG1

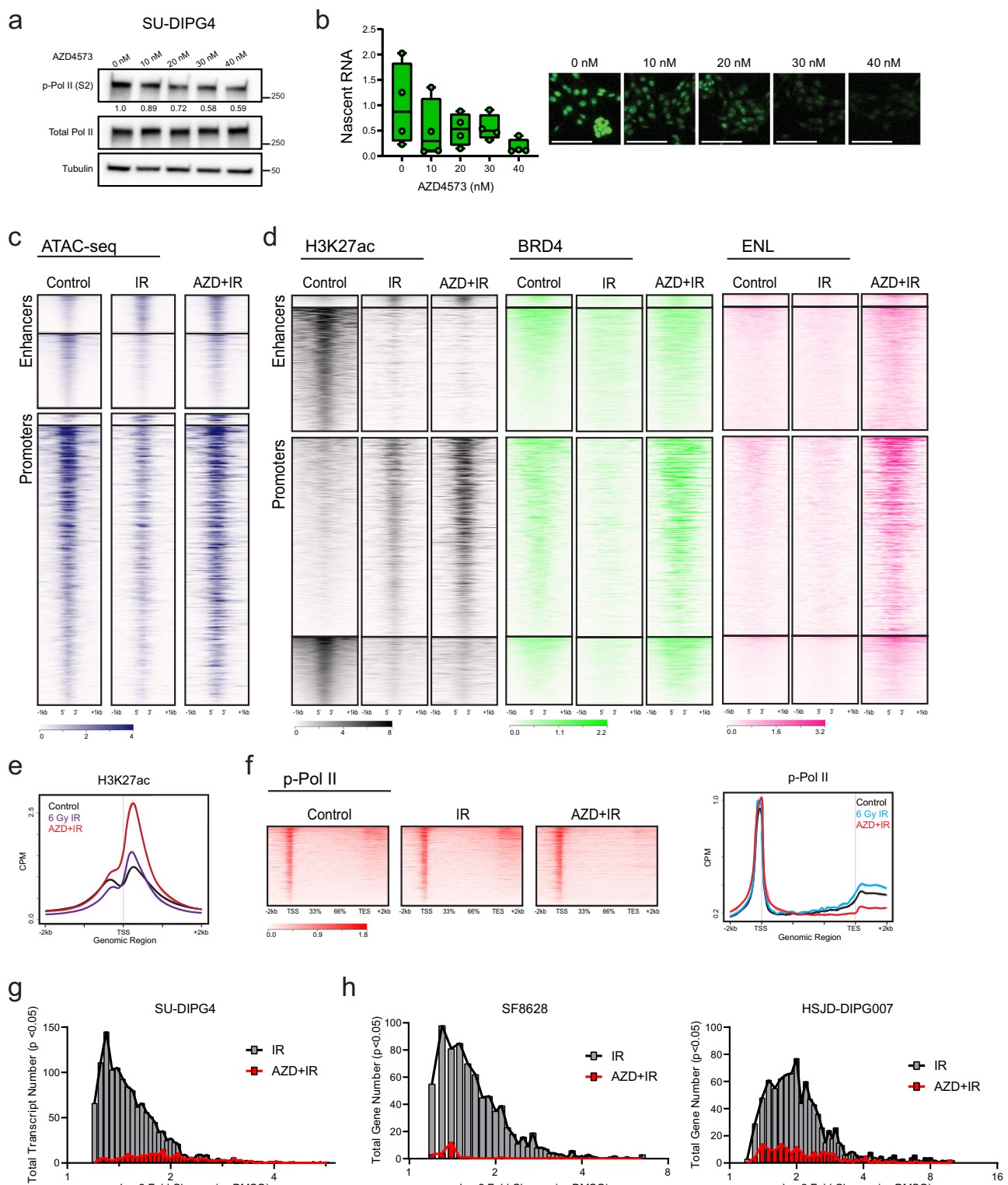

**Fig. 3 | Concurrent CDK9 inhibition disrupts IR-driven chromatin reorganization and abrogates transcriptional induction. a** Immunoblot of p-Pol II (Ser2) and total Pol II at indicated doses of AZD4573. Quantification normalized to total Pol II shown below. Data represent two independent experiments. **b** Click-IT fluorescent assay of relative nascent RNA abundance at indicated doses of AZD4573 (bar = 100 μm), $n = 4$ biologically independent replicates. Box plots display interquartile range, median, and whisker (minimum to maximum). **c** ATAC-seq heatmap of untreated SU-DIPG4 controls, 6 Gy IR exposed, or IR exposed with concurrent AZD4573 (40 nM) ($n = 2$). Genome is clustered by a change in ATAC-seq peaks following IR exposure. **d** H3K27ac, BRD, and ENL heatmap of untreated SU-DIPG4 controls, 6 Gy IR exposed, or IR exposed with concurrent AZD4573 (40 nM) ($n = 2$). Genome is clustered by a change in H3K27ac ChIP-seq peaks at enhancers or promoters following IR exposure. **e.** H3K27ac metagene profile at transcripts differentially upregulated following IR in the presence or absence of AZD4573. **f** p-Pol II heatmap (left) and metagene profile (right) of untreated controls, IR exposed, or IR exposed with concurrent AZD4573 (40 nM) ($n = 3$). **g** Histogram of IR-induced transcripts (ANOVA $p < 0.05$) LFC value in the presence or absence of AZD4573 (SU-DIPG4). **h** IR-induced gene (ANOVA $p < 0.05$) LFC value in the presence or absence of AZD4573 in SF8628 and HSJD-DIPG007 cells. Source data are provided as a Source Data file.

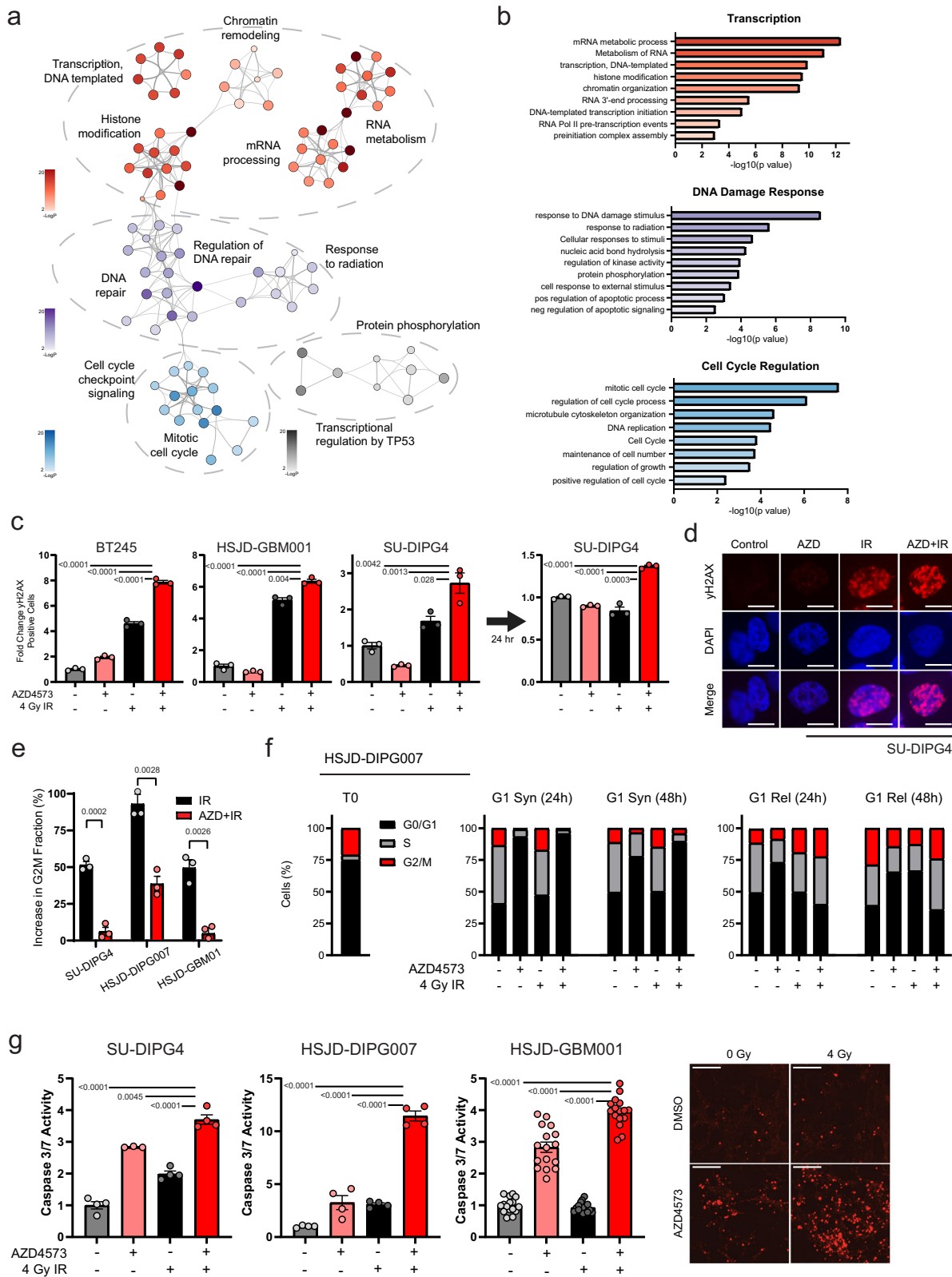

following IR in the presence or absence of AZD4573. We observed an increase in BRG1 phosphorylation by 8 h following IR exposure. Notably, this activation was abrogated in the presence of concurrent CDK9i (Supplementary Fig. 3), correlating with the observed ATAC-seq dynamics. This data would be consistent with a BRG1-mediated mechanism for P-TEFb regulating chromatin compaction, notably active in the early hours following IR.

In contrast to the accessibility changes, we observed a different pattern of response in H3K27ac occupancy with AZD4573 co-treatment. At loci which lost acetylation following IR exposure, little change was seen with P-TEFb inhibition. However, at promoter regions that gained H3K27ac after IR, acetylation paradoxically was markedly increased with AZD4573 co-treatment. Enhancer elements with H3K27ac gains following IR, though small in number, likewise showed a

**Fig. 4 | P-TEFb activity is required for canonical DNA damage response programs.** Gene ontology network (**a**) and terms (**b**) constructed from genes significantly induced by 6 Gy IR but abrogated with concurrent AZD4573 (40 nM) across three cell lines (SU-DIPG4, HSJD-DIPG007, SF8628; *n* = 3 each condition, LFC +/− 1.2 with p val <0.05). Each node denotes an enriched term, with color density reflecting -log(Pval). Enrichment defined by Metascape using hypergeometric test and Benjamini-Hochberg P value correction algorithm. **c**. DNA damage as measured by flow cytometry for γH2AX 6 h (left) or 24 h (right) after IR, 8 nM AZD4573, or combination. Comparison reflects *p* value of two-tailed Student's t-test, mean ± SEM of *n* = 3 biologically independent replicates. **d**. Representative immunofluorescent staining for γH2AX in SU-DIPG4 at same timepoint and conditions as (**c**) (bar = 10 μm). **e**. IR-induced G2M arrest in the presence or absence of

AZD4573 as measured by increase in G2M fraction from cells treated with 8 nM AZD4573, IR, or combination. Comparison reflects *p* value of two-tailed Student's t-test, mean ± SEM of n = 3 biologically independent replicates per cell line. **f**. Cell cycle distribution of HSJD-DIPG007 cells synchronized to G0/G1 (T0, left) and then treated either while in G0/G1 synchronization (G1 Syn, middle) or after G1 release (G1 Rel, right). n = 3 biologically independent replicates per condition. **g**. Caspase 3/7 activation measured 24 h after treatment with 4 nM AZD4573, IR, or combination. Comparison reflects *p* value of two-tailed Student's t-test, mean ± SEM of n = 4 biologically independent replicates, with HSJD-GBM001 imaged 4 fields per replicate. Representative fluorescent live-cell imaging from HSJD-GBM001 shown on right (bar = 200 μm). Source data are provided as a Source Data file.

modest increase in acetylation with co-treatment. Significant increases in both BRD4 and ENL binding were observed accompanying this gain in H3K27ac (Fig. 3d). Given the mechanistic link between these epigenetic cofactors and transcriptional output, we again examined H3K27ac occupancy and p-Pol II elongation, specifically at the TSS of transcripts differentially upregulated following IR. P-TEFb inhibition resulted in a significant accumulation of H3K27ac deposition immediately (<500 bp) downstream from the TSS at these loci (Fig. 3e). Despite this, active transcription from these loci was largely abolished. p-Pol II profiles showed a complete loss of downstream egress with concurrent AZD4573 treatment, instead showing a sharp peak paused at the promoter-proximal region (Fig. 3f). Of the 1,337 unique transcripts differentially upregulated following IR (LFC ≥1.2 qval <0.05), only 230 maintained significant upregulation in the presence of AZD4573 (Fig. 3g and Supplementary Fig. 4a). This created a promoter state in which inhibition of CDK9-mediated pause-release decouples H3K27ac occupancy from productive transcriptional elongation, with the observed hyperacetylation presumably reflecting ineffective epigenetic upregulation accruing behind a stalled Pol II. To test the reproducibility of this phenomenon across different model systems, we performed RNA-seq (n = 3) in the same conditions in additional cell lines (SF8628 and HSJD-DIPG007) representing unique molecular backgrounds (including *H3F3A*, *TP53*, and *ACVR1* status). In each instance, genes with induced expression following IR exposure remained largely quiescent in the presence of AZD4573 co-treatment (Fig. 3h and Supplementary Fig. 4b). Together, these findings outline a model in which H3K27ac-driven recruitment of P-TEFb-containing complexes is required for early Pol II induction following exposure to IR, and that selective inhibition of CDK9 catalytic activity within that window of time largely abrogates this adaptive response.

## P-TEFb activity is required for the early induction of many canonical DNA damage response programs

To characterize the functional consequences of this imparted transcription defect, we next examined the gene expression networks enriched within the RNA-seq datasets (SU-DIPG4, HSJD-DIPG007, and SF8628). Early expression changes following a single IR exposure unsurprisingly showed enrichment in canonical DNA damage response (DDR) programs. Specific genes upregulated showed only modest overlap between models (Supplementary Fig. 5a-b and Supplementary Data 3), perhaps reflecting the genomic heterogeneity between cultures (e.g. mutations to *H3F3A*, *HIST1H3B*, *TP53*, or *ACVR1*) that has been shown to contribute to non-uniformity in radiation response[43,61,62]. Despite this, across the three culture systems, ontology analysis of genes induced by IR exposure but abrogated in the presence of AZD4573 consistently resolved into functional programs broadly involved in transcriptional processing, DNA repair, and cell cycle regulation (Fig. 4a, b and Supplementary Data 4). Alterations in transcriptional processing recapitulated our earlier observations and included enrichment in terms related to chromatin organization, transcription initiation, RNA

metabolism, 3′-end processing, and RNA splicing, with specific perturbation of *CSTF1*, *CDK12*, and mediator subunit expression observed (Supplementary Data 4).

We next validated the predicted deleterious impact of P-TEFb inhibition on the activation of these functional programs. The phosphorylation of the histone variant H2AX can be used as an indirect marker of DNA double-strand breaks (DSBs), the most injurious of IR-induced DNA lesions, which unrepaired can lead to genomic instability and cell death[63]. DSBs can be repaired by error-prone non-homologous end joining (NHEJ) or comparatively error-free by homologous recombination (HR)[64]. When examining the RNA-seq data for canonical marker genes of each repair pathway[65], we observed a significant downregulation of many HR genes (*BARD1*, *BRCA1/2*, *PALB2*, *RAD51*) with comparative sparing of NHEJ genes (*PAXX*, *XRCC4-6*), suggestive of an induced HR defect (Supplementary Table 1 and Supplementary Fig. 6). We then tested accumulation of γH2AX foci in DIPG (SU-DIPG4 and BT245) and HGG (HSJD-GBM001) cultures following IR in the presence or absence of AZD4573. This revealed a reproducible early increase in γH2AX-positive cells in combination-treated cultures relative to control or IR alone. When we repeated this assay 24 h following IR exposure, DSBs in cells treated with IR alone had largely repaired, while γH2AX foci persisted in cells treated with IR in combination with AZD4573 (Fig. 4c, d and Supplementary Fig. 7).

Arrest of the cell cycle at the G2/M checkpoint is a key regulatory mechanism for ensuring genome stability after the detection of IR-induced DNA damage[64,66], and it preferences the HR pathway essential for accurate repair of DSBs[64,65,67]. Consistent with this, cell cycle distribution in DIPG and HGG cultures showed between 48-91% increase in G2/M fraction following a single IR exposure. This protective arrest was largely absent, though, in the presence of concurrent P-TEFb inhibition, with subsequent cell cycle distribution showing little to no change in comparison to untreated controls (Fig. 4e and Supplementary Figs. 7, 8). However, previous reports have described P-TEFb inhibition leading to a G0/G1 arrest[68,69], and G1-arrested cells may not exhibit an appreciable G2 checkpoint. To account for this, we synchronized HSJD-DIPG007 cells in G0/G1 and then treated either before or after G1 release. Cells treated with AZD4573 in G1 did not escape from G1 regardless of IR exposure, while cells treated after G1 release exhibited an intact G2/M arrest in response to IR (Fig. 4f). Taken together, these data suggest that AZD4573 treatment arrests cells at G1, precluding HR programs that, while critical for repair of IR-induced DSBs, are largely restricted to G2[64,65,67]. Consequently, when we examined cell death via induction of caspase 3/7 activity in these conditions, we observed a marked increase in apoptosis when cells were exposed to IR with P-TEFb effectively inhibited (Fig. 4g). Our findings suggest that much of the canonical protective cellular response to IR-induced DNA damage is contingent on an early P-TEFb-mediated transcriptional engagement, and that concurrent CDK9i may cause these critical adaptive programs to collapse.

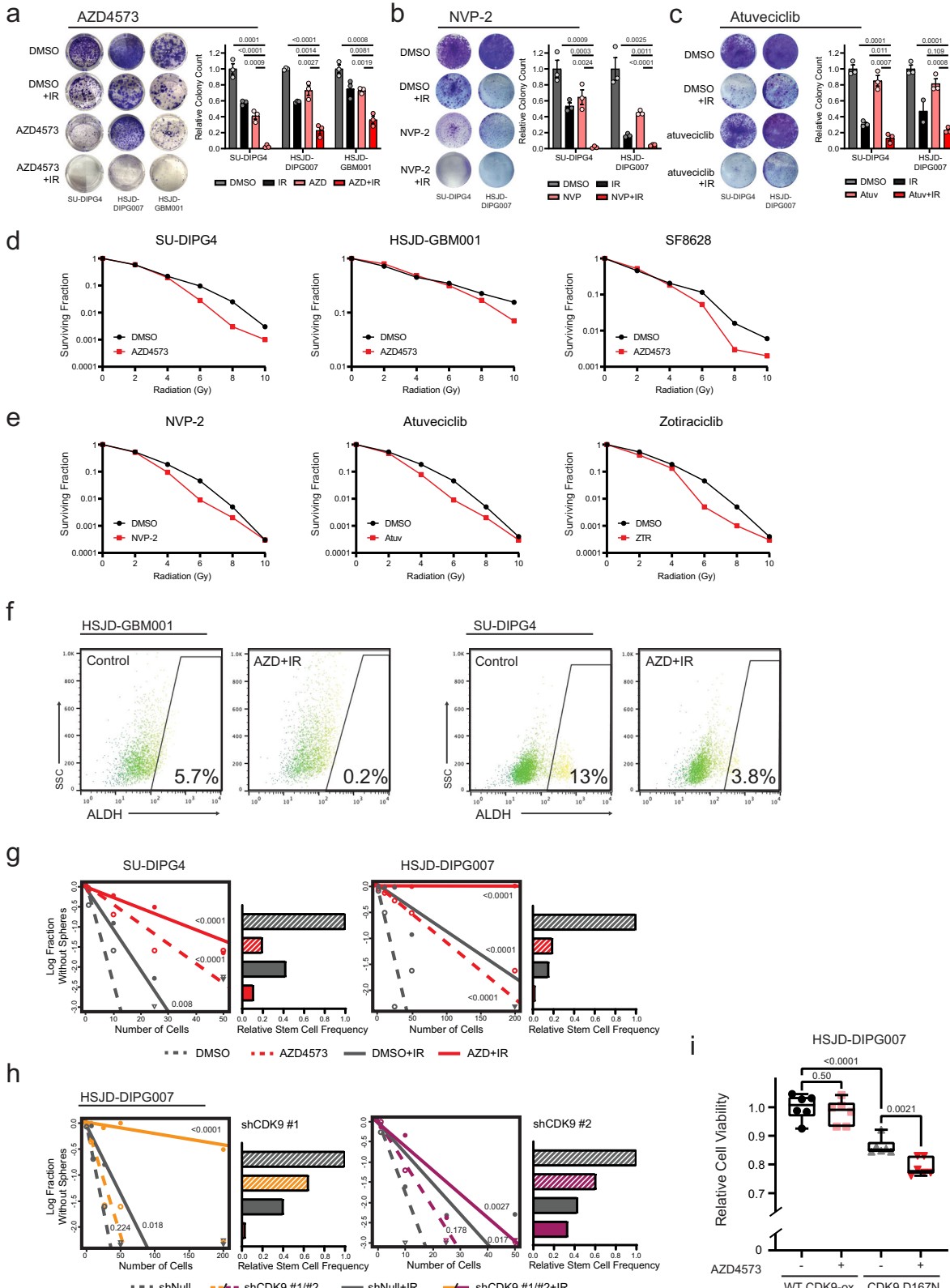

## Concurrent P-TEFb inhibition exhibits cytotoxic synergy with ionizing radiation

Given the observed functional consequences to canonical pro-survival programs, we examined whether CDK9 pharmacologic inhibition could be employed concurrently with IR in order to augment its therapeutic effect. We first sought to optimize the timing of AZD4573 administration around IR exposure. Consistent with an early critical window for IR-adaptive transcriptional response, we found that maximal induction of apoptosis was achieved with treatment just prior to or concurrent with IR exposure; the addition of AZD4573 after IR showed no additional pro-apoptotic effect (Supplementary Fig. 9a, b). Using this approach, clonogenic survival of both DIPG (SU-DIPG4 and HSJD-DIPG007) and HGG (HSJD-GBM001) cultures was significantly diminished by AZD4573 in combination with IR when compared to

**Fig. 5 | CDK9i exhibits cytotoxic synergy with IR in HGG.** Colony focus assay images (left) and quantification (right) of HGG cultures treated with 4 nM AZD4573 (**a**), 6 nM NVP-2 (**b**), or 700 nM atuveciclib (**c**), IR, or in combination. Quantitative comparisons reflect *p* value of two-tailed Student's t-test, mean ± SEM of *n* = 3 biologically independent replicates. **d.** Clonogenic survival for HGG cultures treated with IR alone or combination with 2 nM AZD4573, n = 3 biologically independent replicates. **e.** SU-DIPG4 clonogenic survival treated in combination with IR alone or in combination with 6 nM NVP-2, 700 nM atuveciclib (Atuv), or 15 nM zotiraciclib (ZTR), n = 3 biologically independent replicates. **f.** Brain tumor initiating cell fraction as identified by ALDH expression before and after combinatorial AZD4573 + IR treatment, *n* = 2 biologically independent replicates. **g.** Neurosphere formation efficacy (left) and relative stem cell frequency (right) by extreme limiting dilution assay following treatment with 4 nM AZD4573, IR, or combination.

Comparisons reflect *p* value of pairwise one-sided Chi-square test for stem cell frequencies, data reflect single experiment per cell line with replicates per density in Source Data file. **h.** Neurosphere formation efficacy and relative stem cell frequency by extreme limiting dilution assay following CDK9 shRNA transduction compared to non-targeting control. Insert reflects *p* value of pairwise one-sided Chi-square test, data reflect single experiment per cell line with replicates per density in Source Data file. **i.** Viability of HSJD-DIPG007 cells following 4 Gy IR and ±4 nM AZD4573 treatment. Cells were modified to overexpress either WT CDK9 or D167N catalytic-inactive mutant. Quantitative comparisons reflect *p* value of two-tailed Student's t-test of n = 6 biologically independent replicates, box plots display interquartile range, median, and whisker (minimum to maximum). Source data are provided as a Source Data file.

either intervention alone (Fig. 5a). We observed no difference in sensitivity to P-TEFb inhibition based on histone or *TP53* mutation status (Supplementary Fig. 10a–c and Supplementary Table 2). To strengthen the reproducibility of this therapeutic effect, we repeated these clonogenic assays using NVP-2 and atuveciclib, two additional small molecule inhibitors with high selectivity for CDK9[70]. A similar combinatorial anti-tumor effect with IR was observed (Fig. 5b, c). When cultures were seeded at increasing density and normalized to plating efficiency, a significant radiation dose enhancement effect[71] consistent with synergistic (as opposed to additive) interaction was observed across small molecule CDK9 inhibitors (Fig. 5d, e).

Orthogonal to measurements of gross viability or proliferation is assessment of the brain tumor initiating cell (BTIC) fraction within a tumor population. These stem-like cells are responsible for tumor initiation and have been linked to therapy resistance and cancer regrowth after treatment[72–74]. Aldehyde dehydrogenase (ALDH) has been proposed as a BTIC marker in both pediatric and adult gliomas[75,76]. We quantified ALDH expression after AZD4573, IR, or combination and observed a significant depletion of the ALDH+ cell fraction following combinatorial therapy, greater than was seen either treatment alone (Fig. 5f and Supplementary Fig. 11). We next assessed potential for self-renewal directly through neurosphere extreme limiting dilution assays (ELDA). This demonstrated a reduction in stem cell frequency within all treatment arms but most significantly in combination-treated cells (Fig. 5g). To confirm that this effect was a result of AZD4573's action against CDK9 specifically, we performed parallel experiments using shRNA knockdowns targeting CDK9. This replicated the previously observed reduction in stem cell frequency, with the degree of reduction paralleling shRNA knockdown efficiency (Fig. 5h and Supplementary Fig. 12). Overexpression of wild-type CDK9 rescued cells from the radiosensitizing effect of AZD4573, while overexpression of the catalytically inactive D167N mutant CDK9[77] did not (Fig. 5i). Collectively, these data support the use of CDK9-specific inhibition as a means of augmenting the established therapeutic effect of ionizing radiation.

**Relative transcriptional addiction in HGG supports a therapeutic index for CDK9i in comparison to non-transformed cells**
One could reasonably anticipate that sustained inhibition of transcriptional elongation would be uniformly deleterious across most cellular systems, raising the potential for unacceptable toxicities when adapted for clinical use. Neither of the P-TEFb-member CDK9/CyclinT pair are recurrently mutated or overexpressed in pediatric HGG[40,78,79]. In fact, CDK9 has been characterized as a pan-essential gene by several groups, with a loss of fitness or cell death observed following inhibition in multiple normal tissues or human cell lineages[80–82]. Consistent with this, when we broadened our cell viability assays following 72 h of continuous exposure to AZD4573, we observed no difference in selectivity between neoplastic glioma cultures and normal cell controls (e.g. astrocytes, fibroblasts, or epithelial cells) (Fig. 6a).

Despite this, the pharmacologic targeting of pan-essential genes has formed the core of most successful systemic chemotherapeutic regimens, so long as proper consideration is given to strategies maximizing a therapeutic index between normal tissues and a target cell population of interest[80]. Transcriptional addiction can be defined as an acquired reliance on the continuous activity of an oncogenic transcriptional program[83]. This phenomenon has been identified spanning many cancer types and driver mutations, and it gives rise to specific transcriptional dependencies in cancer cells which are comparatively absent in their non-transformed counterparts[83–88]. We examined both Pol II Ser2 phosphorylation, the primary catalytic output of P-TEFb[21], as well as MCL1 expression, a rapid-turnover anti-apoptotic protein with established P-TEFb-dependence[59], over time following AZD4573 treatment in both glioma and normal astrocyte cultures. At equivalent AZD4573 dosing and timepoints, we observed a more rapid abrogation of P-TEFb catalytic activity and depletion of MCL1 expression in HGG cells when compared to normal astrocytes (Fig. 6b). This correlated with apoptosis as assessed by caspase 3/7 activity on live cell imaging; while all culture systems eventually displayed comparable evidence of cell death, induction was markedly more rapid in neoplastic models in comparison to normal controls (Fig. 6c). Overexpression of MCL1 rescued cells from this effect, delaying caspase 3/7 induction and blunting AZD4573's impact on HGG cell viability (Supplementary Fig. 13a–c). In light of this time-dependent differential sensitivity, we transitioned to a short-term exposure strategy for AZD4573. This was modeled in vitro by drug washout after 8 h, which mimicked the rapid target engagement and established half-life from PK/PD studies in vivo[59]. This strategy revealed a significant difference in sensitivity between neoplastic and non-transformed culture models (Fig. 6d), supportive of a therapeutic index amenable to tolerable intervention.

Normal brain tissue exhibits a relative radioresistance when compared to most glial neoplasms[89,90], a feature exploited daily in the routine clinical treatment of CNS malignancies[91]. Given the differential sensitivity to CDK9i we observed using an intermittent dosing strategy, we examined the comparative tolerance of normal astrocytes to this combinatorial regimen. Using a fixed dose of AZD4573 or atuveciclib and IR, we were able to elicit a significant decrease in DIPG and GBM culture viability while comparatively sparing parallel normal astrocyte and fibroblast controls (Fig. 6e and Supplementary Fig. 14). To further substantiate this therapeutic index, we generated a co-culture system in which DIPG cells (HSJD-DIPG007) and astrocytes (NHA-hTERT) were transfected with GFP- or NucRed-expressing lentivirus, respectively, before plating together at equivalent density. Following fractionated radiotherapy (2 Gy x 3 doses) combined with intermittent AZD4573 administration, we observed a selective depletion of the DIPG fraction relative to normal astrocytes (Fig. 6f, g and Supplementary Fig. 15). Together, these data support the existence of overlapping therapeutic windows in which differential sensitivities to CDK9i and radiation could be exploited to achieve anti-tumor effect while minimizing toxicities to normal CNS tissue.

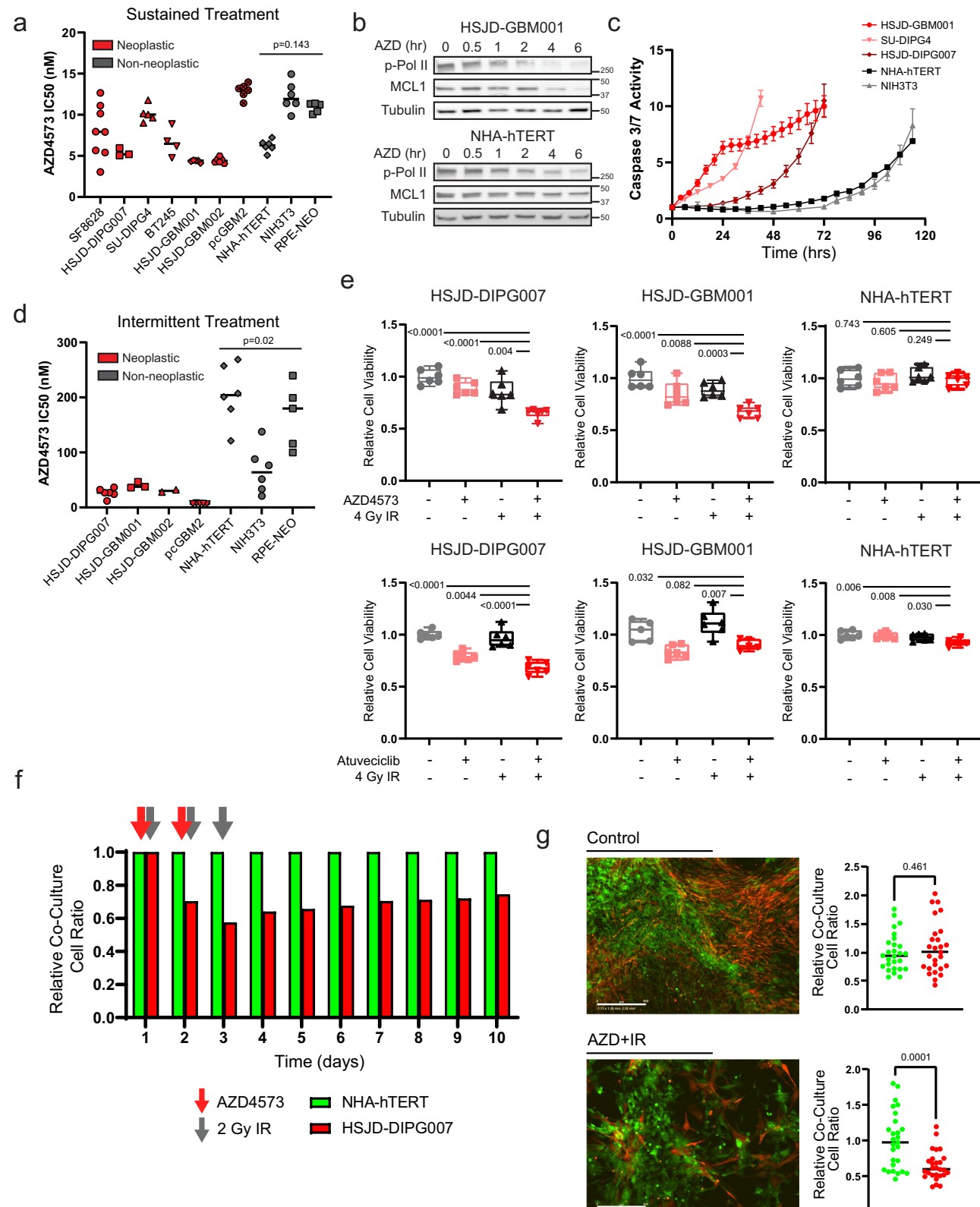

## Effective P-TEFb inhibition augments the anti-tumor effect of radiotherapy in murine models of HGG

Finally, we examined whether this approach of disrupting IR-adaptive transcriptional circuitry could be employed for therapeutic effect in animal models of pediatric high-grade glioma. We utilized SU-DIPG13*, a well-characterized, aggressive model of DIPG, engrafted in the flank of NOD scid gamma mice. Treatment with AZD4573 (15/15 mg/kg

biweekly[59]), either alone or in combination with radiotherapy (2 Gy x 3 doses), resulted in a marked delay of tumor progression and prolongation of survival, including long-term survival in a substantial portion of animals (Fig. 7a–c). Unfortunately, AZD4573 is likely to be a substrate of both the ABCB1 (P-glycoprotein/MDR1) and ABCG2 (breast cancer resistance protein [BCRP]) cell membrane transporters[92,93] based on in vitro transporter assays, with a high efflux

**Fig. 6 | Transcriptional addiction in HGG gives rise to a therapeutic index for CDK9i relative to normal astrocytes. a**. Half-maximal inhibitory concentration of AZD4573 after 3-day exposure in respective cell lines. Comparison reflects $p$ value of two-tailed Student's $t$-test of mean IC50 values from neoplastic vs non-neoplastic cultures. Minimum n = 3 biologically independent replicates per cell line (n = 3 HSJD-DIPG007, n = 4 BT245, n = 5 SU-DIPG4, HSJD-GBM001, and RPE-NEO, n = 6 HSJD-GBM002, pcGBM2, NHA-hTERT, and NIH3T3, n = 8 SF8628). **b**. Western blot analysis of p-Pol II (Ser 2) and MCL1 after indicated exposure times to 50 nM AZD4573. Data represent single experiment. **c**. Caspase 3/7 activity over time following fixed 4 nM dose of AZD4573. Error bars indicate SEM from minimum 4 biological replicates per cell line (n = 4 HSJD-GBM001, NHA-hTERT, and NIH3T3, n = 5 SU-DIPG4 and HSJD-DIPG007). **d**. Half-maximal inhibitory concentrations and comparison as in (A) but measured 3 days after a single 8-hour drug exposure followed by drug washout. Comparison reflects $p$ value of two-tailed Student's $t$-test of mean IC50 values from neoplastic vs non-neoplastic cultures, minimum n = 2 biologically independent replicates per cell line (n = 2 HSJD-GBM002, n = 3 HSJD-GBM001, n = 5 pcGMB2 and RPE-NEO, n = 6 HSJD-DIPG007, NHA-hTERT, and NIH3T3). **e**. Cell viability measured at 3 days following 8-hour exposure to 5 nM AZD4573 (top) or 700 nM atuveciclib (bottom) +/− 4 Gy IR. Box plots display interquartile range, median, and whisker (minimum to maximum). Comparison reflects $p$ value of two-tailed Student's $t$-test, n = 6 biologically independent replicates. **f**. Relative ratio of co-cultured DIPG cells (HSJD-DIPG007) and normal astrocytes (NHA-hTERT) following fractionated radiotherapy and intermittent AZD4573 treatment as indicated by arrows. **g**. Quantification of (f) at day 10, $p$ value of two-tailed Student's $t$-test from n = 6 biologically independent replicates imaged 4 fields per replicate. Source data are provided as a Source Data file.

ratio predicted across the blood-brain barrier (BBB) (AstraZeneca, Investigator Brochure). Consistent with this, the SU-DIPG13* engrafted orthotopically demonstrated no survival benefit from treatment with AZD4573, either alone or in combination with radiotherapy (Supplementary Fig. 16a, b). This supports the therapeutic utility of effective on-target CDK9 inhibition, though it highlights a translational limitation of this specific compound in a CNS disease context.

Zotiraciclib is an orally-available CDK9 inhibitor with established CNS penetrance in several preclinical models[94–96] and an acceptable safety profile in phase I/II trials for adults with anaplastic astrocytoma or glioblastoma[97] (NCT03224104, NCT02942264). Within our culture models of pediatric HGG, zotiraciclib replicated previous CDK9-dependent effects on Pol II phosphorylation, MCL1 depletion, and radiosensitization in vitro (Supplementary Fig. 17). We then tested zotiraciclib (50/35 mg/kg 3x weekly) in combination with fractionated radiotherapy (2 Gy x 3 doses) against the SU-DIPG13* model engrafted orthotopically in the pons (Fig. 7d). This resulted in a decrease in tumor growth by bioluminescent imaging and a significant prolongation of survival in animals treated with combination therapy compared to radiotherapy alone (Fig. 7e–g). To then expand the robustness of this characterization, an additional cohort of the BT245 xenograft model was similarly treated and assessed by MRI. Three-dimensional volumetric analysis of T2-turboRARE MRI sequences showed a significant decrease in tumor size with zotiraciclib treatment compared to IR alone (Fig. 7g and Supplementary Fig. 18). Combination treatment was tolerable, with no animals exhibiting symptomatic, radiographic, or histologic evidence of radionecrosis (Supplementary Figs. 18, 19).

In total, these findings replicate our in vitro observations that effective inhibition of P-TEFb catalytic activity augments the anti-tumor therapeutic effect of IR. While the unique anatomic and physiologic considerations for CNS tumors necessitate the selection of appropriate agents for adequate CNS delivery, our data supports the use of clinically-relevant CDK9 inhibitors in a multimodal treatment approach for these lethal cancers.

## Discussion

This integrative epigenetic and transcriptional analysis provides a framework for understanding the reorganization of transcriptionally active chromatin that underlies the early adaptive response to radiotherapy. Specifically, we show that pHGG cells rapidly compact active chromatin while redistributing H3K27ac deposition in order to modulate transcriptional output in the early hours following an IR exposure. Importantly, we identify that key enzymatic steps in this adaptive cascade are amenable to pharmacologic targeting. Inhibition of the CDK9-catalyzed phosphorylation of the Pol II CTD during the immediate peri-radiation window abrogates not only the induction of critical transcriptional DDR programs, but also BRG1-mediated chromatin reorganization itself (Fig. 8). This inhibition demonstrates a marked anti-tumor effect agnostic of tumor histone, TP53, or passenger mutation status, opening a therapeutic avenue that may prove resilient to the intra- and intertumoral heterogeneity that has complicated translational efforts in a molecularly diverse disease entity.

Pediatric high-grade gliomas are highly lethal malignancies, accounting for the largest proportion of cancer-associated deaths in children[98]. Radiation remains the only uniformly accepted standard of care across HGG subtypes. As such, mechanisms to further sensitize these tumors to radiotherapy, augmenting the depth and duration of clinical response, remain attractive strategies for clinical practice (reviewed elegantly in Metselaar, et al.[62]). Unlike the distributed toxicity potential with combinatorial systemic therapies, the synergistic potential of a radiosensitizing agent is largely confined to the conformal radiation field. A synergistic interaction with standard treatment also lowers practical barriers to clinical translation, as novel radiosensitizers can be more readily incorporated into existing therapy backbones for up-front phase 1 investigation.

Our present study builds on the body of data dissecting the complex reorganization of transcriptional machinery required for the rapid, coordinated induction of large gene expression programs in response to exogenous stressors. This has been best characterized within models of heat shock[30,33,99–102], in which Pol II is released from a paused state into gene bodies for productive elongation across heat shock responsive loci. More recent work has defined considerable specificity within this response. P-TEFb may exit its 7SK-Hexim reservoir and localize to chromatin in several active complexes, including BRD4-P-TEFb, AFF1-SEC, or AFF4-SEC[51,56,103]. Using an auxin-inducible degron system, Zheng et al showed that AFF4-SEC was predominantly responsible for early induction of the heat shock stress response, while other active P-TEFb-containing complexes were largely dispensable[29]. P-TEFb-dependent transcriptional alteration has now also been defined in stress responses to other genotoxic stimuli including chemical carcinogens, albeit with less clearly delineated complex specificity[27]. Our approach here predominantly using short hairpin RNAs or small molecules targeting the CDK9 catalytic component of P-TEFb accepts a tradeoff of molecular specificity for translational potential. Our data lacks the granularity to convincingly define which P-TEFb-containing complex is primarily responsible for the observed IR-induced response, limiting extrapolation to the expected benefit from BET inhibitors[104] or SEC-specific compounds[105,106]. However by targeting the common CDK9-mediated catalytic reaction directly, this strategy bypasses the redundancy observed to some degree with the various P-TEFb-containing complexes[29], ensuring the desired therapeutic endpoint agnostic of a distinct carrier complex.

CDK9 has been recognized as a promising target for cancer therapy for more than a decade, prompting the formulation of numerous inhibitory compounds now in various stages of preclinical and clinical development[107,108]. These have ranged from the first multi-kinase inhibitors such as flavopiridol and dinaciclib to newer, highly-selective agents like AZD4573[59] or intriguing peptidomimetics targeting SEC directly[105]. Early phase I and II clinical trials with flavopiridol and dinaciclib were limited by high rates of adverse events with only

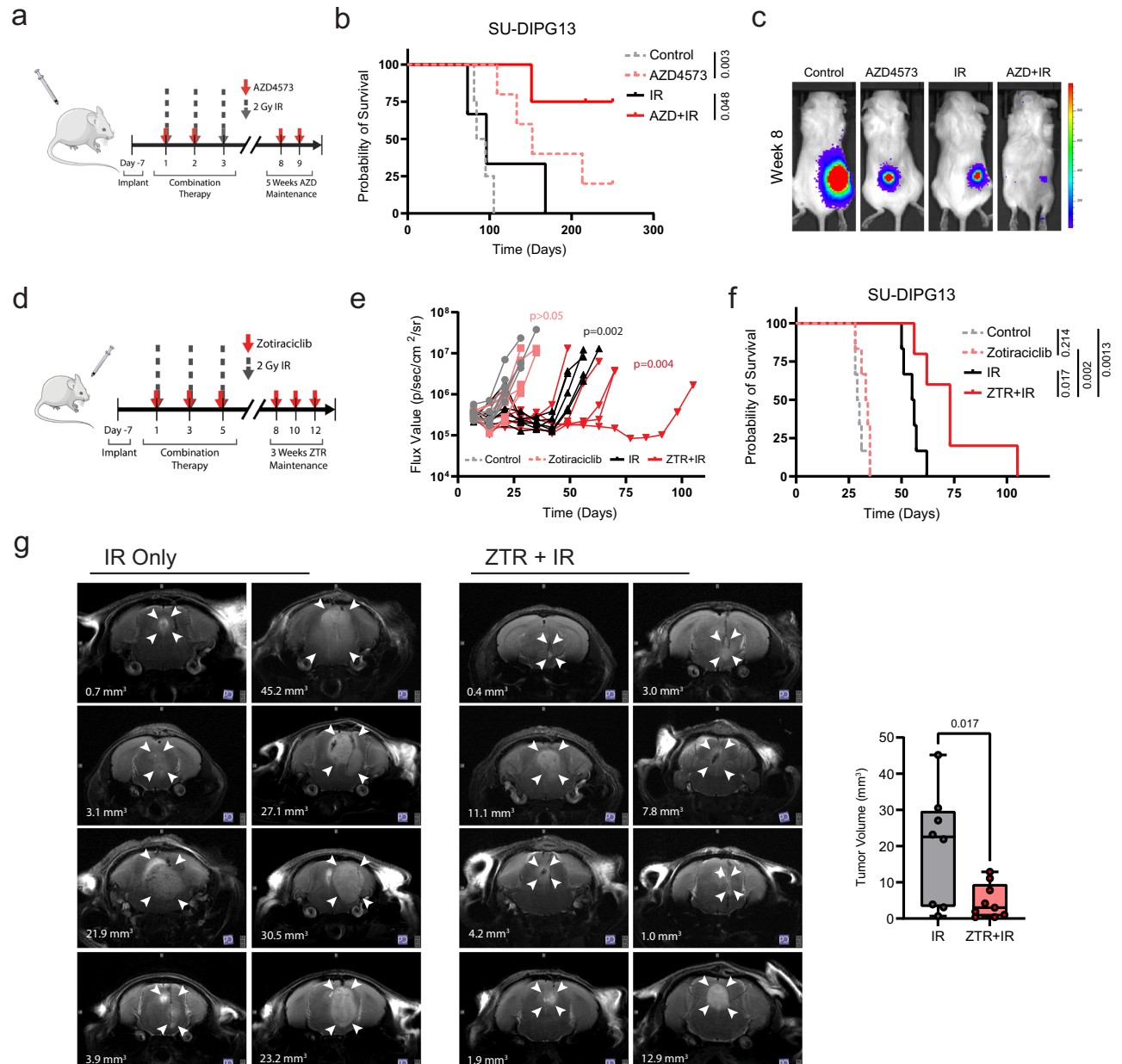

**Fig. 7 | Concurrent CDK9i augments anti-tumor effect of IR to prolong survival in vivo. a**. Schematic represents the treatment schedule of SU-DIPG13* xenografts with either AZD4573 (15/15 mg/kg biweekly administered intraperitoneally), radiotherapy (2 Gy x 3 fractions), or combination. **b**. Kaplan-Meier survival analysis of SU-DIPG13* flank cohorts receiving indicated treatments. Comparison reflects *p* value of Mantel-Cox log-rank test (control n = 4, AZD4573 n = 5, IR n = 3, AZD + IR n = 4). **c**. Bioluminescent imaging from median mouse of each treatment cohort in (B) at completion of therapy period. **d**. Schematic representation of schedule for SU-DIPG13* xenografts treated with either zotiraciclib (ZTR, 50 mg/kg 3x weekly for two weeks followed by 35 mg/kg 3x weekly for two weeks, administered by oral gavage), radiotherapy (2 Gy x 3 fractions), or combination. **e**. Bioluminescent flux

from individual animals within indicated treatment groups (two-tailed Mann-Whitney test at completion of treatment, p value in insert). **f**. Kaplan-Meier survival analysis of orthotopic xenograft cohorts receiving indicated treatments. Comparison reflects *p* value of Mantel-Cox log-rank test (control n = 6, ZTR n = 6, IR n = 6, ZTR + IR n = 5). **g**. Representative axial T2-weighted turboRARE MRI sequences of IR- or ZTR + IR-treated mice. Arrowheads indicate margins of tumor; white text overlay denotes three-dimensional tumor volume. Comparison of tumor volume quantification shown on right (*p* value of two-tailed Student's t-test, IR n = 8, ZTR + IR n = 9). Box plots display interquartile range, median, and whisker (minimum to maximum). Source data are provided as a Source Data file.

modest disease response[109–114], frequently attributed to dose-limiting off-target effects by these relatively non-specific compounds[107]. Phase I/II trials with more selective agents are now ongoing (e.g. NCT03263637, NCT04630756, NCT03224104, NCT02942264). Our data here suggests that not only selectivity but timing and duration of dose exposure is critical in achieving a tolerable therapeutic index, supporting similar observations reported in the preclinical development of AZD4573 for hematologic malignancies[59]. It likewise echoes

what has now been borne out in early clinical trials, in which a phase I dose escalation study of zotiraciclib encountered significant dose limiting toxicities (DLTs) with continuous daily dosing while an intermittent dosing schedule was able to reach the maximum defined dose level without DLTs reported[115]. Furthermore, as many CDK9 inhibitors are under active clinical development primarily for hematologic malignancies or extracranial sarcomas, few compounds have robust data available on effective penetration across the blood brain barrier.

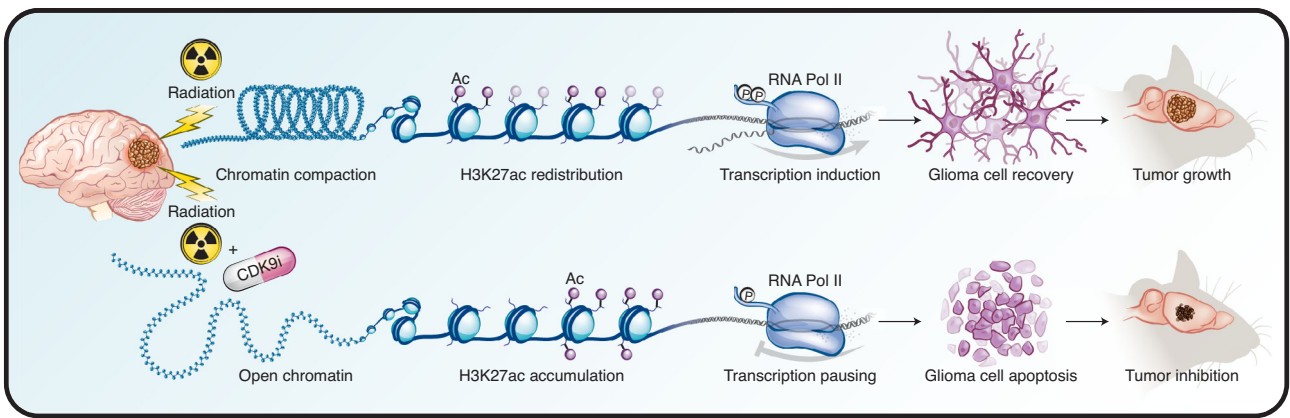

**Fig. 8 | Model of IR-induced transcriptional reorganization.** Rapid reorganization of active chromatin drives transcriptional induction required for DDR programs. Concurrent inhibition of P-TEFb-mediated transcriptional elongation abrogates this adaptive response, augmenting the anti-tumor effect of radiotherapy.

Our data here supports the broader testing of this class of compounds against CNS malignancies.

In conclusion, this study identifies P-TEFb as serving a pivotal role in mediating the transcriptional reorganization required to support an early DNA damage response to radiotherapy. The centrality of this function creates a therapeutic window surrounding IR exposure, in which effective P-TEFb inhibition causes many core adaptive programs to collapse. Judicious dosing strategies can safely exploit this dependency in models of high-grade gliomas, opening avenues of therapy for these recalcitrant malignancies.

## Methods

This research complied with all relevant ethical regulations and the study protocols were approved by the Institutional Biosafety Committee and the Institutional Animal Care and Use Committee at the University of Colorado Anschutz Medical Campus.

### Cell lines and culture technique

HGG and normal cells were maintained as previously described[36,116]. Briefly, SU-DIPG4 and SU-DIPG13* cells were cultured in tumor stem media (TSM) consisting of Neurobasal(-A) (Invitrogen), human-basic FGF (20 ng/mL; Shenandoah Biotech), human-EGF (20 ng/mL; Shenandoah Biotech), human PDGF-AB (20 ng/mL; Shenandoah Biotech), B27(-A) (Invitrogen), and heparin (10 ng/mL). HSJD-DIPG007 cells were maintained in TSM media as above with 10% fetal bovine serum (FBS) (Atlanta Biologicals). BT245 cells were grown in NeuroCult NS-A media (Stemcell Technologies) supplemented with penicillin-streptomycin (1:100, PenStrep)(Gibco/ThermoFisher), heparin (2 μg/mL), human EGF (20 ng/mL), and human FGFb (10 ng/mL). SF8628 cells were maintained in Dulbecco's modified eagle medium (DMEM)(Gibco/ThermoFisher) and supplemented with 10% FBS, MEM non-essential amino acids (Gibco/ThermoFisher) and antibiotic/antimicotic (Gibco/ThermoFisher). Normal human astrocytes were cultured in DMEM and supplemented with 10% FBS, PenStrep, L-glutamine (Gibco/ThermoFisher), and sodium pyruvate (Gibco/ThermoFisher). NIH3T3 cells were cultured in DMEM and supplemented with 10% FBS and PenStrep. Cells were grown in either adherent monolayer conditions (Falcon/Corning) or as tumor neurospheres (ultra-low attachment flasks, Corning) as indicated. All cell lines were validated by DNA fingerprinting through the University of Colorado Molecular Biology Service Center utilizing the STR DNA Profiling PowerPlex-16 HS Kit (DC2101, Promega)(Supplementary Data 5).

### Plasmid and lentiviral production

HEK293FT cells were used to produce viral particles by packaging PSPAX2 and PMD2.G vectors with shCDK9 plasmids purchased from the University of Colorado Functional Genomics Core Facility (TRCN0000000494 and TRCN0000199780).MCL1 over-expression (clone ID: ccsbBroad304_00985) and empty vector plasmids were similarly purchased from the University of Colorado Functional Genomics Core Facility. Wild-type (pBABE-Flag-Cdk9-IRES-eGFP) and D167N mutant CDK9 (pBABE-Flag-Cdk9-D167N-IRES-eGFP) plasmids were as described by Dow et al.[77] and purchased from addgene (USA).

### Sensitivity enhancement ratios

Cells were seeded in triplicate 1000 to 10,000 cells in 6 well plates. They were treated with zero to 10 Gy at 2 Gy increments in the presence or absence of 3 nM AZD4573. After 14 days, media was aspirated, cells were washed with 1x PBS, and stained with crystal violet for 15 min before being manually counted.

### Cell viability assays

Cells were treated with AZD4573 (AstraZeneca), TG02 (MedChemExpress), or ionizing radiation as indicated before adding MTS reagent (Promega) and measuring absorbance at 492 nM on a BioTek Synergy H1 microplate reader (BioTek Instruments, VT).

### Caspase 3/7 assays

Indicated cells were seeded at a density of 4000 cells per well on 96 well plates. Radiation was administered at 4 Gy, immediately followed by the addition of caspase 3/7 dye (Sartorius) at 0.5uM. Cells were monitored over time using IncuCyte S3 Live Cell Analysis System, with red or green reporter count normalized against time zero values.

### Aldehyde dehydrogenase assay

ALDH activity of was measured using Aldefluor kit (Stem Cell Technologies) according to the manufacturer's instruction. Cells were stained with propidium iodide and then analyzed on the Guava easyCyte HT flow cytometer (Luminex).

### Extreme limiting dilution assay

Limiting dilution assays were performed as we[36] and others[117] have previously described. Briefly, cells were seeded on a 96-well ultra-low-attachment round-bottom tissue culture plate in serum-free media at increasing numbers from 1 cell/well to 100 cells/well. Cells were allowed to grow for 14 days, and the number of wells containing neurospheres was counted manually under light microscopy. Published ELDA software (http://bioinf.wehi.edu.au/software/elda/) was used to calculate comparative self-renewal potential of cells.

## H2AX flow cytometry

Cells were seeded and radiated in the presence or absence of AZD4753. At least 1,000,000 cells were collected per condition and washed with 2% FBS in 1x PBS wash buffer. The DMSO control was split into two samples to be used as a negative control. Cells were spun and resuspended in 100uL of BD CytoFix Fixation buffer (554655) for 10 min at room temperature. Samples were spun, washed, and incubated with -20-degree BD Phosflow permeation buffer (558050) for 5 min at room temperature. After incubation, cells were spun and washed again and resuspended in Alexa Fluor 488.

## Click-iT nascent RNA imaging

Cells were plated at 50% confluency in a 96-well microplate. The following day, cells were irradiated at 4 Gy and proceeded to the reaction immediately after the radiation, 1 hr, 2 hrs, 4 hrs, and 8 hrs after radiation, or treated with 0, 10, 20, 30, or 40 nM of AZD4573 for 1 h. Reactions were performed accordingly to the manufacturer instructions (Click-iT RNA HCS Assay, C10327, Invitrogen). Briefly, cells were treated with 1 mM of EU and incubated under normal cell conditions for 1 h. After EU incubation, media was removed and 50 ul of 3.7% formaldehyde in PBS in each well was added and incubated for 15 min at room temperature. The fixative was then removed, and cells were washed once with PBS. 50 ul of 0.5% Triton X-100 in PBS was added and incubated for 15 min at room temperature. Cells were washed once with PBS, and 50 ul of Click-iT cocktail was added and incubated for 30 min at room temperature, protected from light. After the incubation, Click-iT cocktail was removed and cells were washed once with 50 ul of Click-iT reaction rinse buffer. Cells were washed with PBS and then incubated with 50 ul of HCS NuclearMask Blue stain solution, diluted 1:1000 in PBS, and incubated for 15 min at room temperature, protected from light. HCS NuclearMask Blue stain solution was removed and cells were washed with PBS twice. Fluorescence was then quantified using the Incucyte S3 (Sartorius).

## Cell cycle analysis

Cells were seeded 500,000 per 100 mm dish and treated at 70 - 80% confluence with DMSO, 8 nM AZD5473, 4 Gy radiation, or combination. Cells were collected at 24 h, fixed with 1-3 mL ice cold 70% EtOH and stored at -20 degrees overnight. The pellet was washed in a wash buffer of 2% FBS in 1x PBS, spun, and resuspended in 100uL of fluorescent nuclear dye DRAQ5, diluted 1:1000 in wash buffer. The suspensions were incubated for 30 min at 37-degrees in the dark. After incubation, cells were run on an Amnis FlowSight cytometer equipped with a 488 nm laser.

For synchronization, cells were grown to 40-50% confluence and then serum starved with 0% FBS for 72 h before reintroducing 10% FBS. Cells were treated under the same conditions either at the time of serum reintroduction or 24 hr post serum reintroduction. Cells were then collected at 24, 48, and 72 h, fixed with 1-3 mL ice cold 70% EtOH and stored at -20 degrees overnight prior to analysis as per above.

## Co-Immunoprecipitation

Immunoprecipitation experiments were performed with SU-DIPG4 whole cell extracts. Cell lysates were immunoprecipitated using Universal MagneticCoIPKit (ActiveMotif 54002), then analyzed by western blot. 500 ug of whole cell extract per sample was precleared for 5 min on ice using 20 uL of protein G magnetic beads. Supernatant was then immunoprecipitated using 7 ug anti-CDK9 antibody (Cell Signaling, [C12F7] #2316) or rabbit IgG. Antibody/extract mixtures were incubated with complete Co-IP/wash buffer on a 4-degree shaker for 4 h. Magnetic protein G beads were added and incubated for another hour, then washed using the complete co-IP/wash buffer and a magnetic stand. Bead pellets were re-suspended in 2X reducing loading buffer for immunoblotting.

## Western blotting

Whole-cell protein lysates were harvested in lysis buffer (RIPA buffer supplemented with protease inhibitor (Roche), sodium vanadate and sodium molybdate) from cells in indicated conditions. Protein was separated on 4-20% PROTEAN TGX Gels and blotted using a wet transfer system (Biorad) before probing for CDK9 (Abcam [EPR3119Y] (ab76320)), phospho-Rbp1 CTD (Ser2) (Cell Signaling (E1Z3G) #13499), phospho-Rbp1 CTD (Ser5) (Cell Signaling (D9N5I) #13523 S), LARP7 (abcam # ab134746), p-BRG1 (Cell Signaling [Ser1627/1631] [E2N9V] #58034), or α-Tubulin (Cell Signaling (DM1A) #3873 S).

## Immunofluorescence

Cells were plated in 8-well chamber and 24 h later were submitted to the following treatments: DMSO, 8 nM AZD, IR, or combination. IR was given 2 hr after drug treatment. Reaction for gH2AX (BD Pharmigen; Alexa Fluor 488 Mouse anti-H2AX (Ps139); 560445) was performed 6hs after radiation; reaction for CDK9 (Cell Signaling, (C12F7) #2316 S) was performed after 4 hrs. Cells were washed in PBS, fixed with 4% paraformaldehyde for 20 min, and washed 3 times with PBS for 5 min each. Cell permeabilization was performed using Triton X-100 0.3% for 5 min and unspecific reaction was blocked incubating cells with 3% BSA for 1 h at room temperature. Cells were incubated with a primary antibody diluted 1:100 in 3% BSA and incubated overnight at 4 C. For anti-H2AX reactions, cells were washed with PBS 3 times and slides were mounted with coverslips using the ProLong Gold antifade reagent with DAPI (Invitrogen; P36935). For anti-CDK9 reactions, cells were washed 3 times with PBS and secondary antibodies were applied diluted 1:1000 in PBS and incubated for 1 hr at room temperature, protected from light. Cells were washed with PBS 3 times, slides were mounted with coverslips using the ProLong Gold antifade reagent with DAPI (Invitrogen; P36935), and images were taken using the a confocal microscope (Keyence).

## Transcriptome sequencing (RNA-seq)

Ribonucleic acid was isolated from cells in indicated experimental conditions using a Qiagen miRNAeasy kit (Valencia, CA). Illumina Novaseq 6000 libraries were prepared and sequenced by the Genomics and Microarray Core Facility at the University of Colorado Anschutz Medical Campus. Resulting sequences were filtered and trimmed, removing low-quality bases (Phred score <15), and analyzed using a custom computational pipeline consisting of gSNAP for mapping to the human genome (hg38), expression (FPKM) derived by Cufflinks, and differential expression analyzed with ANOVA in R. Output files contained read-depth data and FPKM expression levels for each sample, and when gene expression levels were compared between groups of samples, the ratio of expression in log2 format and a P value for each gene was recorded. Subsequent gene ontology analysis was performed by hypergeometric test and Benjamini-Hochberg P value correction algorithm using the Metascape platform[118].

## Assay for transposase accessibility to chromatin (ATAC-seq)

Cells were plated in a 10 cm-plate and the following day were treated with DMSO, 40 nM AZD4573, 6 Gy IR, or combination. IR was administered 2 h after drug treatment. 4 h after IR, cells were scraped and counted. 100,000 cells were spun down at 500 x g for 5 min at 4 °C, pellet was washed once with 500 ul of cold 1x PBS, and spun down at 500 x g, 5 min at 4 °C. Pellet was resuspended in 450 ul of cold hypotonic buffer (10 mM Tris-HCl, pH 7.4, 10 mM NaCl, 3 mM MgCl$_2$) and immediately added 50 ul of 1% IGEPAL CA-630 (0.1% final) and inverted to mix. Cells were incubated on ice for 15 min, spun down at 500 x g for 10 min at 4 °C. Pellet was set on ice, and the transposition reaction was performed following the manufacturer instructions (Cell Biologics, Cat No. CB6936). Pelleted cells were resuspended in 50 ul of transposition reaction mix (25 ul 2X Reaction Buffer, 2.5ul Transposome, 22.5ul

Nuclease free water) and incubated at 37 °C for 1 h. DNA was purified using Qiagen MinElute kit (Qiagen, #28004) and eluted in 15 ul of Elution Buffer, followed by the Library Generation to amplify transposed DNA fragments. For this reaction the following reagents were mixed in a PCR tube, 10ul of Transposed DNA, 10 ul of Nuclease free water, 2.5 ul of Ad1.noMX (Oligo 1), 2.5 ul Ad2. Barcode (Oligo 2), and 25 ul of High Fidelity 2x PCR Master Mix. This mixture was run on a thermocycler using the following cycle: 98 °C 30 sec, (98 °C 10 sec, 63 °C 30 sec, 72 °C 1 min) x 10 times and held at 4 °C. For double-sided bead purification (to remove primer dimers and large >1000 bp fragments), each PCR sample was transferred to a 1.5 ml tube and 0.5X volume (25 ul) of AMPure XP beads was added, mixed by pipetting 10 times, and incubated at room temperature for 10 min. Tubes were placed in a magnetic rack for 5 min, and the supernatant was transferred to a new tube. 1.3X of the original volume (65 ul) of AMPure XP beads was added, mixed by pipetting 10x, and incubated at room temperature for 10 min. Tubes were placed in a magnetic rack for 5 min, supernatant was discarded, and beads were washed with 200 ul 80% EtOH. EtOH was removed and tubes were left on magnetic rack with cap open for 10 min or until the pellet was totally dry. Beads were resuspended in 20ul of nuclease-free water and mixed thoroughly, tubes were placed in the magnetic rack for 5 min, and the supernatant was transferred to a new tube. Libraries were paired-end sequenced on NovaSEQ 6000 platform.

### Chromatin immunoprecipitation (ChIP-seq)

Cells were crosslinked with 1% formaldehyde added to the growth media for 10 min, followed by quenching with 0.125 M glycine for 5 min. Cells were washed twice with cold PBS, collected in ice-cold PBS by scraping, pelleted, and resuspended in cell lysis buffer (5 mM PIPES, pH 8.0; 85 mM KCl; 0.5% NP-40). Following incubation on ice for 10 min, a nuclear-enriched fraction was collected by centrifugation for 5 min at 2500 x g at 4 °C. Pellet was resuspended in ChIP lysis buffer (50 mM Tris-HCl, pH 8.0; 10 mM EDTA; 1%SDS) containing COmplete EDTA free protease inhibitors (Sigma), and quantified with BCA. 2 ug/ul of lysates were sonicated with a Bioruptor Plus (Diagenode) for 20 cycles (30 seconds ON and 30 seconds OFF). The size of the sonicated DNA fragments was checked on an 1X TAE 1% agarose gel electrophoresis and ranged between 250 to 500 bp. After removing the debris via centrifugation, chromatin extracts were collected.

For immunoprecipitation, chromatin extracts were diluted with Chip Dilution Buffer (16.7 mM Tris-HCl, pH8.1, 1.2Mm EDTA, 167 mM NaCl, 0.01%SDS and 1.1% Triton x100) and incubated with primary antibodies (anti-H3K27Ac, Active Motif #39133) overnight at 4 °C. After incubation with the primary antibody, 20 uL of pre-washed magnetic beads (Magna ChIP Protein A + G Magnetic Beads, Millipore Sigma) were added to each sample for 2 h at 4 °C. Using a magnetic rack, the immunoprecipitates were washed successively with 1 ml of low salt buffer (20 mM Tris-HCl [pH 8.0], 150 mM NaCl, 0.1% SDS, 1% triton X-100, 2 mM EDTA), high salt buffer (20 mM Tris-HCl [pH 8.0], 500 mM NaCl, 0.1% SDS, 1% triton X-100, 2 mM EDTA), LiCl washing buffer (10 mM Tris-HCl [pH 8.0], 250 mM LiCl, 1.0% NP40, 1.0% deoxycholate, 1 mM EDTA) and twice with TE buffer. The DNA-protein complexes were eluted with 300 μl of IP elution buffer (1% SDS, 0.1 M NaHCO3). The cross-links were reversed by adding NaCl (a final concentration 0.2 M) into the eluents and incubating them at 65 °C overnight. The DNA was recovered by proteinase K and RNase A digestion, followed by phenol/chloroform extraction and ethanol precipitation. Pellets were resuspended in 25ul of RNase and DNase free water, and ChIP-DNA was quantified using the Qubit dsDNA High Sensitivity Assay kit (Thermo Fisher Scientific).

### Cleavage under target and release using nuclease (CUT&RUN)

Beads were prepared using 10 ul/sample of CUTANA Concanavalin A Conjugated Paramagnetic Beads (EpiCypher, SKU:21-1401). Beads were transferred to a 1.5 ml tube and placed in a magnetic separation rack, supernatant was removed, and beads were washed 2 times with 100 ul/sample of cold Bead Activation Buffer (20 mM HEPES pH7.9, 10 mM KCl 1 mM CaCl2, 1 mM MnCl2). Beads were then resuspended in 10 ul/sample of cold Bead Activation Buffer, and aliquot 10 ul/sample of activated bead slurry into 8-strip tube and kept on ice.

Cells were collected using 0.05% trypsin, washed with 100 ul/sample of Wash Buffer (20 mM HEPES pH7.5, 150 mM NaCl, 0.5 mM Spermidine, 1x Roche cOmplete EDTA-free Protease Inhibitor (1187358001)) at room temperature, and centrifuged at 600 g for 3 min, for a total of two washes. Cells were resuspended in 100 ul/sample RT Wash Buffer, and then an aliquot of 100 ul washed cells was added to each 8-strip tube containing 10 ul of activated beads and mixed by pipetting. Cells were incubated with bead slurry for 10 min at RT. Tube strip was placed on a magnetic separation rack until slurry clears and the supernatant was removed. 50 ul of cold Antibody Buffer was added to each sample and mixed by pipetting; 1ul of antibody IgG (Rabbit IgG Negative Control, EpiCypher, 13-0042 K), BRD4 (EpiCypher, 13-2003), ENL (Cell Signaling, [D9M4B] #14893), or p-Pol II (pRpb1 CTD (S2), Cell Signaling, #13499 S), was added to each sample, mixed and incubated on a nutator overnight at 4 °C. Tube was placed on a magnet until slurry clears, and the supernatant was removed. While beads were on magnet, 250 ul of cold Digitonin Buffer was added directly onto beads of each sample and then pipetted to remove supernatant, for a total of 2 washes. 50 ul of cold Digitonin Buffer was added to each sample and mixed. 2.5ul of CUTANA pAG-MNase (20Xstock) was added to each sample, mixed, and incubated for 10 min at RT. Tube was placed on a magnet and supernatant was removed. Beads were washed 2 times with cold Digitonin Buffer. Supernatant was removed, and 50 ul of cold Digitonin Buffer was added to each sample and mixed. Tubes were placed on ice, and 1 ul of 100 mM CaCl2 was added to each sample and mixed, and then incubated on nutator 2 hrs at 4 °C. 33 ul/sample of Stop buffer containing 0.5 ng/sample of E.coli spike-in DNA, was added to the samples, mixed by pipetting, and incubated for 10 min at 37 °C in a thermocycler. Tubes were placed on a magnet stand until slurry cleared, and the supernatant was transferred to a 1.5 ml tube. Finally, the DNA was purified using CUTANA DNA Purification Kit (EpiCypher, 14-0050) according to the manufacturer instructions, and 1 ul of DNA was used for quantification by Qubit.

### Library preparation

The NEBNext Ultra DNA Library Prep Kit for Illumina (NEB #E7645S) with Dual Index Primers (NEB #E7600) were used for library preparation with ChIP-seq.

**NEBNext end prep.** 1x TE was added to the DNA to bring final volume to 50 ul and mixed with 3 ul of NEBNext Ultra II End Prep Enzyme Mix and 7 ul of NEBNext Ultra II End Prep Reaction Buffer. This was placed in thermocycler with heated lid set to 75 °C and run 30 min at 20 °C, 60 min at 50 °C, and held at 4 °C.

**Adaptor ligation.** The adaptor ligation reaction was performed with 60 ul of Prep Reaction Mixture, 2.5 ul of NEBNext Adaptor for Illumina diluted following the manufacture recommendations, 30 ul of NEBNext Ultra Ligation Master Mix, and 1 ul of NEBNext Ligation Enhancer. This was incubated at 20 °C for 15 min with heated lid off. 3 ul of USER Enzyme was added to the reaction and incubated at 37 °C for 15 min with heated led set to 47 °C.

**Cleanup of adaptor-ligated DNA without size selection.** 1.1x of AMPure XP Beads were added to the Adaptor Ligation reaction, and samples were incubated for 5 min at room temperature. Tubes were placed on a magnetic stand to separate beads from supernatant, and after 5 min, supernatant was removed and beads were washed 2 times

with 200 ul of 80% freshy prepared ethanol. Beads were air dried for up 5 min while tubes were on magnet with lid open. DNA was eluted from the beads by adding 17ul of 0.1XTE buffer.

**PCR enrichment of adaptor-ligated DNA.** To 15 ul of the Adaptor Ligated DNA fragments was added 25 ul of NEBNext Ultra Q5 Master Mix, 5 ul of Index primer i7, and 5 ul Universal PCR Primer i5. PCR amplification was performed following the cycling conditions: 98 °C for 45 s, 98 °C 15 s and 60 °C 10 s for 16 cycles, 72 °C for 1 min, and hold at 4 °C. Clean up of the PCR reaction was performed using AMPure XP Beads.

### Sequencing Analysis
The quality of the fastq files was accessed using FastQC (v.0.11.8)[119] and MultiQC[120]. Illumina adapters and low-quality reads were filtered out using BBDuk (http://jgi.doe.gov/data-and-tools/bb-tools). Bowtie2 (v.2.3.4.3)[121] was used to align the sequencing reads to the hg38 reference human genome. Samtools (v.1.11)[122] was used to select the mapped reads (samtools view -b - q 30) and sort the bam files. PCR duplicates were removed using Picard MarkDuplicates tool (http://broadinstitute.github.io/picard/). The normalization ratio for each sample was calculated by dividing the number of uniquely mapped human reads of the sample with the lowest number of reads by the number of uniquely mapped human reads of each sample. These normalization ratios were used to randomly sub-sample reads to obtain the same number of reads for each sample using using samtools view -s. Bedtools genomecov was used to create bedgraph files from the bam files[123]. Bigwig files were created using deepTools bamCoverage[124] and visualized using IGV[125]. Peaks were called using MACS2 (v2.1.2)[126] using ENCODE recommendations. IDR was used to identify the reproducible peaks between the replicates[127]. Further processing of the peak data was performed in R, using in particular the following tools: valR[128], and DiffBind[129]. Average profiles and heatmaps were generated using ngs.plot[130].

### Orthotopic and flank xenograft models
Animal models were generated as we have previously described[36]. Briefly, NOD SCID gamma (NSG) mice (Jackson Labs) between 6-8 weeks of age were anesthetized and immobilized on a Kopf Model 940 Stereotaxic Frame with a Model 923 mouse gas anesthesia head holder and Kopf 940 Digital Display. To target the pons, a 1.0 mm diameter burr was drilled in the cranium using a Dremel drill outfitted with a dental drill bit at 1.000 mm to the right and 0.800 mm posterior to lambda, with cell suspension injected 5.000 mm ventral to the surface of the skull. A suspension of 100,000 SU-DIPG13 or BT245 cells/2 μl/injection was then slowly injected using an UltraMicroPump III and a Micro4 Controller (World Precision Instruments). Post-surgical pain was controlled with SQ carprofen. For flank model generation, $5 \times 10^7$ SU-DIPG13 cells/200 ul serum-free media are manually injected subcutaneously into mouse flank. Equal numbers of male and female mice were used for both orthotopic and flank xenograft models. Animals were euthanized upon demonstrating neurologic symptoms or weight loss (orthotopic) or tumor size ≥2 cm (flank). University of Colorado Institutional Animal Care and Use Committee (IACUC) approval was obtained and maintained throughout the conduct of the study.

### Animal treatment
Mice were randomized sequentially at 6-8 days following injection. AZD4573 was prepared to 10% DMSO stock and diluted in 40% sterile water, 39% polyethylene glycol 400, and 1% Tween 80. Mice were treated with 15/15 mg/kg IP biweekly[59], either alone or in combination with radiotherapy (2 Gy x 3 doses) for total of 6 cycles or until reaching protocol endpoint. Zotiraciclib (TG02) was prepared to 10% DMSO stock and diluted in 0.5% methylcellulose and 1% Tween 80 in sterile water. Mice were treated with 50 mg/kg three times weekly for two weeks followed by 35 mg/kg three times weekly for two weeks, either alone or in combination with radiotherapy. Conformal radiation was administered as previously described[131]. Briefly, each mouse was anesthetized and positioned in the prone orientation aligned to the isocenter in two orthogonal planes by fluoroscopy. Each side of the mouse brain received half of the dose, which was delivered in opposing, lateral beams. Dosimetric calculation was done using a Monte-Carlo simulation in SmART-ATP (SmART Scientific Solutions B.V.) for the fourth ventricle, mid brain, and pons receiving the prescribed dose. Treatment was administered using a XRAD SmART irradiator (Precision X-Ray) using a 225 kV photon beam with 0.3 mm Cu filtration through a circular 10-mm diameter collimator.

### Animal imaging
Non-invasive magnetic resonance imaging was performed through the University of Colorado Anschutz Medical Campus Animal Imaging Shared Resource as previously described[36]. Imaging was obtained between 6 and 8 weeks post-injection. Scans were performed on an ultra-high field Bruker 9.4 Tesla BioSpec MR scanner (Bruker Medical, Billerica, MA) equipped with a mouse head-array RF cryo-coil. Non-gadolinium multi-sequential MRI protocol was applied to acquire (i) high-resolution 3D T2-weighted turboRARE; (ii) sagittal FLAIR; and (iii) axial fast spin echo DWI. All MRI acquisitions and image analysis were performed using Bruker ParaVision 360NEO software. All MRI acquisitions and image analyses were performed by a radiologist blinded to the treatment assignment of the mice.

### Quantification and statistical analysis
Unless indicated otherwise in the figure legend, all in vitro data are presented as mean ± SEM. For quantitative comparisons, significance was defined as $p \leq 0.05$, * $p = <0.05$, ** $p = <0.01$, *** $p = <0.001$, **** $p = <0.0001$. All statistical analysis was performed on GraphPad Prism 9.0 software (GraphPad, La Jolla, CA). Kaplan-Meier survival curve comparisons were performed by log-rank (Mantel-Cox) test using GraphPad Prism 9.0 software.

### Reporting summary
Further information on research design is available in the Nature Portfolio Reporting Summary linked to this article.

## Data availability
The accession number for the raw and processed data reported in this paper is GEO: GSE227797. Source data are provided as a Source Data file. The remaining data are available within the Article, Supplementary Information, or Source Data file. Source data are provided with this paper.

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

## Acknowledgements

We would like to thank the following individuals for generously providing the cell lines used for this work: Michelle Monje (Stanford University) for SU-DIPG4 and SU-DIPG13, Angel Montero Carcaboso (Sant Joan de Déu) for HSJD-DIPG007, HSJD-GBM001, and HSJD-GBM002, Siddhartha Mitra (Stanford University, University of Colorado) for SU-pcGBM2, and Nalin Gupta (University of California, San Francisco) for SF8268 and SF7761. We likewise thank the University of Colorado Functional Genomics Facility, Genomics and Microarray Shared Resource, Research Histology Shared Resource, and Animal Imaging Shared Resource for their contributions to this work. This work was generously supported by grants through the St. Baldrick's Foundation (Fellowship Award 606095, N.A.D. and R.V.), the National Institute of Child Health and Human Development (5K12HD068372-09, N.A.D.), the National Institute of Neurological Disorders and Stroke (1K08NS121592, N.A.D.), the Morgan Adams Foundation (N.A.D., S.V., and R.V.), the Andrew McDonough B+ Foundation (N.A.D.), the Uncle Kory Foundation (N.A.D.), and the Luke Morin Family (N.A.D., S.V., and R.V.). The University of Colorado Bioinformatics and Biostatistics and Research Histology Shared Resources are supported by a Cancer Center Support Grant (P30CA046934). The University of Colorado Animal Imaging Shared Resource is supported by the P30CA046934 and NIH S10OD023485 grants (N.J.S). S.D.K. is funded National Institute of Dental and Craniofacial Research to SDK (1R01DE028282-01, 1R01DE028529-01) and by 1P50CA261605-01. AZD4573 and TG02 were provided at no cost by AstraZeneca and Cothera Bioscience, respectively, under preclinical MTA. Original scientific illustrations by Katie Vicari. Other figures were generated in part using Servier Medical Art (https://smart.servier.com/) under a Creative Commons Attribution 4.0 unported license.

## Author contributions

L.M.S. performed the ChIP-seq, CUT&RUN, and ATAC-seq experiments. E.D. performed the ChIP-/ATAC-seq and CUT&RUN alignment and

analysis. F.M.W. performed the RNA-seq experiments. B.S. and D.W. performed RNA-seq alignment and analysis. L.M.S., F.M.W., and S.D. performed flow cytometry and in vitro phenotypic assays. A.P. and F.M.W performed the stereotactic xenograft injections; F.M.W. and I.B. conducted the subsequent in vivo studies. S.D.K. designed and administered the in vivo radiotherapy plan. S.K. and N.J.S. performed the xenograft MR imaging and were responsible for its analysis. N.K.F. procured tissue specimens and maintained IRB compliance. F.M.W., L.M.S., and N.A.D. prepared the figures and wrote the manuscript. S.V., N.K.F., R.D., R.V., and N.A.D. conceived the project, supervised all aspects of the work, and edited the manuscript.

## Competing interests

S.D.K received clinical trial funding from Genentech, AstraZenca, and Ionis unrelated to this work. She also receives preclinical funding from Roche, unrelated to this work. The remaining Authors declare no competing interests.
