## [Peer Review File · Nature Communications]

REVIEWER COMMENTS

Reviewer #1 (Remarks to the Author):

The authors investigate the role of PTEFb in the DNA damage response of pediatric high-grade gliomas (HGGs) and its potential as a therapeutic target. The authors find that PTEFb activity is required for the early induction of DNA damage response programs and that inhibition of PTEFb leads to alterations in transcriptional processing, DNA repair, and cell cycle regulation. They also show that PTEFb inhibition exhibits cytotoxic synergy with ionizing radiation and can be used to selectively target HGGs while sparing normal astrocytes. Finally, they demonstrate that effective PTEFb inhibition using AZD4573 and zotiraciclib augments the anti-tumor effect of radiotherapy in flank and orthotopic murine models of HGG.

This study is well executed, presented with appropriate controls and well supported by quantifications. The main finding that pTEFB-mediated transcription regulates cell cycle effects and DNA repair is established and recently reviewed (Anshabo et al., 2021, *Frontiers in Oncology*). The other main finding that DNA damage impacts global changes to 3D genome organization including increased segregation of topologically associating domains (TADs) was also recently reported in *Nature Communications* (Sanders et al., 2020), *Nature* (Arnould et al., 2021) and *Nature Structural & Molecular Biology* (Zagelbaum et al., 2022). The specific link between pTEFB and IR-induced chromatin reorganization is novel and should be of major interest to the scientific community. However, several points need to be addressed to strengthen the validity of the findings reported.

1. The authors show that exposure to radiation caused an increase in PTEFb activity leading to the transcriptional activation of a set of genes that promote the survival of glioma cells in response to radiation. They posit the mechanism of PTEFb recruitment is via H3K27ac. The observation that H3K27ac deposition correlates with differential transcript levels is interesting. However, deeper mechanistic insight would significantly strengthen the study including elucidating which PTEFb binding partners are required. As the authors note, PTEFb may localize to chromatin in several complexes including BRD4-PTEFb, AFF1-SEC, or AFF4-SEC.

2. In the treatment of DIPB, acute side effects of radiotherapy to the brainstem are of critical concern. The authors report differential sensitivity to CDK9 inhibitors and radiation using DIPB cultures with astrocytes and fibroblasts serving as normal CNS tissue controls. There is no in vitro assessment of combined modality therapy toxicity using human neuronal cell lines, neural-derived organoids, or the cortical tissue in vivo. This significantly limits the translational impact of the study.

3. The authors suggest that PTEFb is important for DNA repair. Based on the experiments provided it is unclear whether pretreatment with AZD4573 affects the number DSBs generated by IR or their repair (Fig 4d). It would significantly strengthen the study if the authors imaged gamma H2AX at multiple time points and used assays that distinguish between the efficiency of cNHEJ and HDR with AZD4573 treatment. These experiments would optimally distinguish between cyclin T and cyclin K given published reports that CDK9-cyclin K plays a direct role in the repair of damaged DNA by interacting with components of the ATR pathway.

4. The authors suggest that AZD4573 abolishes the G2/M checkpoint leading to increased apoptosis in cells exposed to IR. However, Fig 4e appears to show G1 arrest in cells treated with AZD4573 alone including the HSJD-GBM01 cell line and to a lesser extent SU-DIPG4 cell line. This finding is congruent with previously reported roles of P-TEFB and BRD4 inhibition causing G1 arrest and apoptosis in the literature. Cells that are arrested in G1 will not exhibit a G2/M checkpoint, and the data as represented in Fig 4f may be an oversimplification. More careful analysis of the cell cycle effects in the setting of AZD4573 would help clarify. Alternatively, repeating experiments on cells synchronized in different phases of the cell cycle, where different repair pathways are favored, would better clarify the relationship between DNA repair, cell cycle distribution, and apoptosis.

5. The authors suggest that ionizing radiation causes a redistribution of H3K27ac at gene promoters allowing for PTEFb to drive Pol II pause-release. Do the authors believe that PTEFb inhibitors impact chromatin compartment switches or TAD boundaries? The authors have already correlated RNA-seq data with ATAC-seq, H3K27ac, and p-Pol II datasets. Hi-C would allow for more direct assessment of changes in 3D interactions between chromatin regions in the setting of AZD4573 treatment.

Reviewer #2 (Remarks to the Author):

Cellular adaptation to stress relies on the dynamic regulation of gene expression. One such regulatory mechanism described in the literature is the simulation of Pol II pause-release at a fraction of genes by P-TEFb kinase. Activation of P-TEFb facilitates the survival of stressed cells, and the targeting of P-TEFb in combination with genotoxic therapies has been proposed (i.e. in PMID 2482433). However, the potential of countering P-TEFb in cancer therapy has not been explored. The study by Walker et al. addresses this important question using pediatric high-grade glioma models. The authors demonstrate that therapeutic ionizing radiation triggers rapid genome-wide reorganization of active chromatin, most notably increased compaction of chromatin at gene promoters, which in turn prompts gene induction in P-TEFb-dependent fashion. Critically, inhibiting P-TEFb disrupts this chromatin reorganization, blunts transcriptional induction, and interferes with key adaptive programs such as DNA damage repair and cell cycle regulation. The authors went on to show that the combination of radiation and P-TEFb inhibition

exhibits potent synergistic therapeutic potential, regardless of glioma subtype, inducing significant tumor cell apoptosis and prolonging xenograft survival.

Conceptually, the significance of active P-TEFb in cell stress is not novel. To my recollection, however, the authors are first to report it in the context of IR. Furthermore, the role of P-TEFb in the regulation of chromatin compaction has been reported previously (PMCID: PMC6247954). Curiously, the authors of that study found that there was a strong correlation between chromatin relaxation and gene expression in response to CDK9 inhibition, but the prolonged treatment with the CDK9i could account for the difference with the present study. Nevertheless, this is a well-designed and timely study that emphasizes the central role of P-TEFb in the early adaptive response to radiotherapy. Critically, it suggests new avenues for combinatorial treatments in these deadly malignancies. Before publication of this work, the authors need to address the comments below.

Major comments:

1. Previous work from i.e. Svejstrup (PMCID: PMC5332558), Barboric (PMCID: PMC6482433) and Zhou (DOI: 10.1038/35104575) groups has shown that genotoxic stress stimulates Pol II pause-release at short genes, and that this is mediated by activation of P-TEFb via the release of the kinase from the inhibitory 7SK snRNP. Fig.2 and Fig.3 gene induction findings of the present study are consistent with the notion that P-TEFb activation plays a vital role in irradiated cells. However, the authors never addressed this question. Ser2 phosphorylation data is provided, but this could be mediated by other CTD kinases, such as CDK12 or CDK13. To strengthen the P-TEFb connection, the authors would need to show that P-TEFb is indeed more active in the irradiated cells, i.e. by in vitro kinase assays or coIP experiments that would monitor the potential release of P-TEFb subunits from 7SK snRNP.

2. Could the authors relate their chromatin compaction findings to the published work on the topic (PMCID: PMC6247954)?

3. Although the authors knocked down CDK9 in one of their functional cell line assays, it would be very important to 1) use an alternative CDK9i in their key functional cell line assays, such as highly selective NVP-2; 2) to control for CDK9 specificity, it would be very useful to employ another highly selective inhibitor against a transcriptional CDK, such as YKL-5-124 (CDK7i) or THZ531 (CDK12/13i). This is important as i.e. CDK12 has been described to play a role in DDR (first published in PMCID: PMC3205586).

4. The authors replicated only a part of their findings using zotiraciclib. The addition of Western blotting data (as in Fig.6b) and Caspase3/7 data would strengthen their argument.

Minor comments:

1. The correct acronym for Positive transcription elongation factor b is P-TEFb, not PTEFb.
2. The references in #1 above should be discussed in the introduction section.
3. Could the authors explain in the text why is SU-DIPG-IV a good cell line model of glioma?
4. Could the authors include the concentrations of drugs used in all of their Figure legends? At present, these are mostly non-existent.
5. For the reader, the legend in Fig.2e could be very useful.
6. Fig.3a: is quantification shown normalized to total Pol II levels?
7. 'Given the role of H3K27ac in recruiting P-TEFb...' – could the authors include the reference to back up this claim?
8. Fig.3c: the clusters shown need a better explanation.
9. Fig.3f: a metagene plot would be very helpful here.
10. Fig.4c: BT245 isn't mentioned in the text that refers to this panel.
11. Fig.4d: Could you label the cell line used for these experiments?
12. Fig.5b: Could the authors quantify the synergy (i.e. Bliss score)?

13. Fig.5c and Extended Fig.6: This needs a better description and presentation of the data, particularly ExtFig.6.

14. ExtFig.7: To appreciate the differences, the images need a better contrast.

15. It would be very interesting to see BETi or SEC-specific compounds in the orthotopic model.

16. As IR has been reported to deplete LARP7 via BRCA1-mediated protein degradation that in turn affects cellular DDR (DOI: 10.1016/j.celrep.2020.107974), the authors need to 1) test whether IR depletes LARP7 in their systems, and if so, 2) discuss how this result relates to their key findings.

Reviewer #3 (Remarks to the Author):

The manuscript entitled "Rapid PTEFb-dependent transcriptional reorganization underpins the glioma adaptive response to radiotherapy" highlights a potential novel therapeutic strategy against diffuse intrinsic pontine gliomas. Overall, the manuscript is executed well. Certain points require improvement:

1. Figure 5E requires a cDNA rescue with wt-CDK9 as well as with an inactive mutant.

2. Fig 6B: They should over-express Mcl-1 to demonstrate functional involvement in apoptosis induction in the related context.

3. Why do they use a DIPG model in the flank? This does not seem to be an appropriate micro-environment for those types of tumors. Probably, this experiment should be included in the supplement.

4. Fig 7G: Why are only two images shown?

Reviewer #1:

The authors investigate the role of PTEFb in the DNA damage response of pediatric high-grade gliomas (HGGs) and its potential as a therapeutic target. The authors find that PTEFb activity is required for the early induction of DNA damage response programs and that inhibition of PTEFb leads to alterations in transcriptional processing, DNA repair, and cell cycle regulation. They also show that PTEFb inhibition exhibits cytotoxic synergy with ionizing radiation and can be used to selectively target HGGs while sparing normal astrocytes. Finally, they demonstrate that effective PTEFb inhibition using AZD4573 and zotiraciclib augments the anti-tumor effect of radiotherapy in flank and orthotopic murine models of HGG.

This study is well executed, presented with appropriate controls and well supported by quantifications. The main finding that pTEFB-mediated transcription regulates cell cycle effects and DNA repair is established and recently reviewed (Anshabo et al., 2021, *Frontiers in Oncology*). The other main finding that DNA damage impacts global changes to 3D genome organization including increased segregation of topologically associating domains (TADs) was also recently reported in *Nature Communications* (Sanders et al., 2020), *Nature* (Arnould et al., 2021) and *Nature Structural & Molecular Biology* (Zagelbaum et al., 2022). The specific link between pTEFB and IR-induced chromatin reorganization is novel and should be of major interest to the scientific community. However, several points need to be addressed to strengthen the validity of the findings reported.

We greatly appreciate the reviewer's characterization of the study as well executed and of major interest to the scientific community. Specific points addressed below.

1. The authors show that exposure to radiation caused an increase in PTEFb activity leading to the transcriptional activation of a set of genes that promote the survival of glioma cells in response to radiation. They posit the mechanism of PTEFb recruitment is via H3K27ac. The observation that H3K27ac deposition correlates with differential transcript levels is interesting. However, deeper mechanistic insight would significantly strengthen the study including elucidating which PTEFb binding partners are required. As the authors note, PTEFb may localize to chromatin in several complexes including BRD4-PTEFb, AFF1-SEC, or AFF4-SEC.

To address this, we have performed additional CUT&RUN experiments probing BRD4 and ENL (SEC family member) in the same experimental conditions. We found that BRD4 binding is significantly depleted following radiation, reminiscent of the overall slowing of observed transcriptional rates, and that this depletion occurs even at the pro-survival genes upregulated following radiation. ENL binding, by contrast, is relatively preserved, particularly at our defined upregulated pro-survival genes. From this, we conclude that the transcriptional activation is a predominantly SEC-PTEFb driven mechanism rather than BRD4-SEC. These data are now included in Figure 2e, 2h, 3d, and Extended Data Figure 2.

2. In the treatment of DIPB, acute side effects of radiotherapy to the brainstem are of critical concern. The authors report differential sensitivity to CDK9 inhibitors and radiation using DIPB cultures with astrocytes and fibroblasts serving as normal CNS tissue controls. There is no in vitro assessment of combined modality therapy toxicity using human neuronal cell lines, neural-derived organoids, or the cortical tissue in vivo. This significantly limits the translational impact of the study.

We acknowledge and appreciate the potential for on-target, off-tumor toxicity with CDK9 inhibition either alone or in combination with radiation. We have made significant efforts to address this, both through detailed in vitro assays (Figure 6a-g and Extended Data Figures 13 and 14) and tolerability across multiple in vivo cohorts (Figure 7 and Extended Data Figure 15). We have now added an additional supplemental figure showing absence of caspase activation or histologic necrosis within the surrounding normal pons from the in vivo study (Extended Data Figure 17). While testing can always

be done in additional cell lines or models (ie neural-derived organoids), this is not an accepted standard for mechanistic preclinical testing across DIPG or similar models in comparable high-impact work (see Golbourn et al *Nat Can* 35422502, Grasso et al *Nat Med* 25939062, Krug et al *Can Cell* 31085178, or Panditharatna et al *Can Discov* 36305736).

3. The authors suggest that PTEFb is important for DNA repair. Based on the experiments provided it is unclear whether pretreatment with AZD4573 affects the number DSBs generated by IR or their repair (Fig 4d). It would significantly strengthen the study if the authors imaged gamma H2AX at multiple time points and used assays that distinguish between the efficiency of cNHEJ and HDR with AZD4573 treatment. These experiments would optimally distinguish between cyclin T and cyclin K given published reports that CDK9-cyclin K plays a direct role in the repair of damaged DNA by interacting with components of the ATR pathway.

As the experiments shown in Figure 4c-d are performed within hours (6 hr) of a single DNA damaging event (IR exposure), we clearly show a significant initial increase in the number of DSBs (or rather γ H2AX foci) generated after pretreatment. We have now assessed repair by persistence of γ H2AX at 24 hours, at which point the foci from IR-alone samples had returned to baseline controls. Combination-treated samples remained modestly elevated above IR-alone (Figure 4c), consistent with an impaired DNA repair process. To evaluate NHEJ vs HR, we specifically examined expression of canonical NHEJ and HR genes from the ChIP- and RNA-seq data. We noted a significant reduction in many core HR genes including *PALB2*, *BRCA1/2*, *BARD1*, and *RAD51* with AZD4573 treatment. By contrast, NHEJ genes (*XRCC4-6*, *PAXX*) were largely unaffected. This supports a predominantly HR effect, and this data has now been included in Extended Data Figure 6 and discussed in the corresponding Results subsection. This HR defect is also consistent with the more nuanced cell cycle analysis, discussed below.

4. The authors suggest that AZD4573 abolishes the G2/M checkpoint leading to increased apoptosis in cells exposed to IR. However, Fig 4e appears to show G1 arrest in cells treated with AZD4573 alone including the HSJD-GBM01 cell line and to a lesser extent SU-DIPG4 cell line. This finding is congruent with previously reported roles of P-TEFB and BRD4 inhibition causing G1 arrest and apoptosis in the literature. Cells that are arrested in G1 will not exhibit a G2/M checkpoint, and the data as represented in Fig 4f may be an oversimplification. More careful analysis of the cell cycle effects in the setting of AZD4573 would help clarify. Alternatively, repeating experiments on cells synchronized in different phases of the cell cycle, where different repair pathways are favored, would better clarify the relationship between DNA repair, cell cycle distribution, and apoptosis.

This is a particularly incisive question and raises an alternate interpretation for our data; we ultimately believe the interpretation suggested by reviewer here to be correct. We synchronized cells in G0/G1 and then treated, both prior to cell cycle entry and after allowing for G1 escape. These data are now included in Figure 4f and described as follows:

“However, previous reports have described P-TEFb inhibition leading to a G0/G1 arrest^{67,68}, and G1-arrested cells may not exhibit an appreciable G2 checkpoint. To account for this, we synchronized HSJD-DIPG007 cells in G0/G1 and then treated either before or after G1 release. Cells treated with AZD4573 in G1 did not escape from G1 regardless of IR exposure, while cells treated after G1 release exhibited an intact G2/M arrest in response to IR (**Figure 4f**). Taken together, these data suggest that AZD4573 treatment arrests cells at G1, precluding HR programs that, while critical for repair of IR-induced DSBs, are largely restricted to G2^{63,64,66}.”

5. The authors suggest that ionizing radiation causes a redistribution of H3K27ac at gene promoters allowing for PTEFb to drive Pol II pause-release. Do the authors believe that PTEFb inhibitors impact chromatin compartment switches or TAD boundaries? The authors have already correlated RNA-seq data with ATAC-seq, H3K27ac, and p-Pol II datasets. Hi-C would allow for more direct assessment of changes in 3D

interactions between chromatin regions in the setting of AZD4573 treatment.

This is an intriguing suggestion and the subject of active ongoing follow-up work. However, the complexity of this characterization is far beyond the technical and narrative scope of the present manuscript.

Reviewer #2:

Cellular adaptation to stress relies on the dynamic regulation of gene expression. One such regulatory mechanism described in the literature is the simulation of Pol II pause-release at a fraction of genes by P-TEFb kinase. Activation of P-TEFb facilitates the survival of stressed cells, and the targeting of P-TEFb in combination with genotoxic therapies has been proposed (i.e. in PMID PMC6482433). However, the potential of countering P-TEFb in cancer therapy has not been explored. The study by Walker et al. addresses this important question using pediatric high-grade glioma models. The authors demonstrate that therapeutic ionizing radiation triggers rapid genome-wide reorganization of active chromatin, most notably increased compaction of chromatin at gene promoters, which in turn prompts gene induction in P-TEFb-dependent fashion. Critically, inhibiting P-TEFb disrupts this chromatin reorganization, blunts transcriptional induction, and interferes with key adaptive programs such as DNA damage repair and cell cycle regulation. The authors went on to show that the combination of radiation and P-TEFb inhibition exhibits potent synergistic therapeutic potential, regardless of glioma subtype, inducing significant tumor cell apoptosis and prolonging xenograft survival.

Conceptually, the significance of active P-TEFb in cell stress is not novel. To my recollection, however, the authors are first to report it in the context of IR. Furthermore, the role of P-TEFb in the regulation of chromatin compaction has been reported previously (PMCID: PMC6247954). Curiously, the authors of that study found that there was a strong correlation between chromatin relaxation and gene expression in response to CDK9 inhibition, but the prolonged treatment with the CDK9i could account for the difference with the present study. Nevertheless, this is a well-designed and timely study that emphasizes the central role of P-TEFb in the early adaptive response to radiotherapy. Critically, it suggests new avenues for combinatorial treatments in these deadly malignancies. Before publication of this work, the authors need to address the comments below.

We appreciate this reviewer's characterization of the manuscript as well-designed, timely, and addressing an important question. We agree with the assessment that P-TEFb significance in cell stress is not novel, but that this is the first report to define its relevance in the context of therapeutic IR. Specific points addressed below.

Major comments:

1. Previous work from i.e. Svejstrup (PMCID: PMC5332558), Barboric (PMCID PMC6482433) and Zhou (DOI: 10.1038/35104575) groups has shown that genotoxic stress stimulates Pol II pause-release at short genes, and that this is mediated by activation of P-TEFb via the release of the kinase from the inhibitory 7SK snRNP. Fig.2 and Fig.3 gene induction findings of the present study are consistent with the notion that P-TEFb activation plays a vital role in irradiated cells. However, the authors never addressed this question. Ser2 phosphorylation data is provided, but this could be mediated by other CTD kinases, such as CDK12 or CDK13. To strengthen the P-TEFb connection, the authors would need to show that P-TEFb is indeed more active in the irradiated cells, i.e. by in vitro kinase assays or coIP experiments that would monitor the potential release of P-TEFb subunits from 7SK snRNP.

This critique raises both an important mechanistic question as well as an opportunity for clarification on our part. To the first, does IR stimulate release of P-TEFb from the inactive 7SK snRNP complex? To address this, we have performed co-IP of CDK9 from cells before and after IR exposure, and probed for the 7SK complex member LARP7. We observed a clear decrease in LARP7 bound to CDK9, and this

data is now included in Figure 2d (there is unfortunately some high background in the IgG lane, but no specific band is seen). To the second, whether P-TEFb is more active following radiation perhaps reflects a misstatement of interpretation on our part. Our data in Figure 2a, 2c, and 2g suggests that Ser2 phosphorylation and net transcription are in fact decreased following IR, consistent with prior work showing (UV) radiation slows transcription rates (Munoz et al 19450518). However, we simultaneously demonstrate SEC binding and increased p-Pol II pause-release at genes essential for early DDR response. Our interpretation of this data is that while global processive transcription is decreased following IR, the residual P-TEFb activity is critical for stimulating important pro-survival responses. To clarify this, we have added the following language:

“Prior work has demonstrated that DNA damaging events transiently decrease elongation rates, phenocopying slow Pol II mutants^{9,46}. This slowdown has functional implications, as Pol II elongation kinetics can alter mRNA co-transcriptional processing, alternative splicing, and alternative isoform expression^{9,46,47}. Consistent with this, in situ fluorescent staining of nascent RNA synthesis showed a time-dependent decrease in net transcriptional output following a single IR exposure (**Figure 2a**). While this slowing enables pro-survival functions such as repair of Pol II-encountered DNA lesions and limiting mutagenesis, it is held in opposition with the need to rapidly activate transcriptional programs involved in early DDR^{8,9}.

...

Regardless, these data demonstrate a central role for H3K27ac redistribution in organizing P-TEFb-driven transcriptional output, including focal induction of critical DDR programs, within the broad chromatin compaction and transcriptional slowing observed following IR-induced genotoxic stress.”

Direct measurement of CDK9 activity (vs CDK7/12/13) is challenging, as to our knowledge the available commercial assays (Promega, amsbio, etc) measure ADP production from purified kinase activity. These are designed for profiling inhibitory compounds and not the dominant mode of regulation by 7SK snRNP seen in vivo. As orthogonal ways of strengthening the P-TEFb connection (vs CDK7/12/13), we have performed the additional CUT&RUN experiments for BRD4 vs SEC as suggested by Reviewer #1 (see above and Figure 2-3), tested additional CDK9-specific compounds as described below (Figure 5), as well as the CDK9 overexpression experiments suggested by Reviewer #3 (see below and Figure 5h). Each of these data are consistent with a P-TEFb-mediated dependency.

2. Could the authors relate their chromatin compaction findings to the published work on the topic (PMCID: PMC6247954)?

We genuinely thank you for this suggestion. This is an illuminating context to consider the ATAC-seq findings in that we had overlooked. We performed Western blot for the phosphorylated Ser1627/1631 positions of BRG1 following IR in the presence or absence of AZD4573. This showed an increase in p-BRG1 following IR, which was abrogated by CDK9 inhibition. We have therefor added the following language:

“This might reflect a direct effect of processive transcriptional activity on local chromatin architecture. Alternatively, prior work has shown that independent of its canonical role in modifying the Pol II CTD, P-TEFb indirectly modifies chromatin compaction via phosphorylation of the chromatin remodeling SWI/SNF complex member protein BRG1⁵⁹. In that study, inhibition of CDK9 activity led to a loss of BRG1 phosphorylation and a subsequent chromatin relaxation in a BRG1-dependent manner⁵⁹. To explore this as a mechanistic explanation for the observed accessibility changes, we performed immunoblot for phospho-BRG1 (Ser1627/1631) following IR in the presence or absence of AZD4573. We observed an increase in BRG1 phosphorylation by 8 hours following IR exposure. Notably, this

activation was abrogated in the presence of concurrent CDK9i (**Extended Data Figure 3**), correlating with the observed ATAC-seq dynamics. This data would be consistent with a BRG1-mediated mechanism for P-TEFb regulating chromatin compaction, notably active in the early hours following IR.”

3. Although the authors knocked down CDK9 in one of their functional cell line assays, it would be very important to 1) use an alternative CDK9i in their key functional cell line assays, such as highly selective NVP-2; 2) to control for CDK9 specificity, it would be very useful to employ another highly selective inhibitor against a transcriptional CDK, such as YKL-5-124 (CDK7i) or THZ531 (CDK12/13i). This is important as i.e. CDK12 has been described to play a role in DDR (first published in PMID: PMC3205586).

To strengthen the P-TEFb/CDK9 specificity, we have performed additional CDK9 overexpression experiments with catalytic mutants (see below and Figure 5h). As suggested here, we have also replicated key functional assays using zotiraciclib as well as NVP-2 and atuvaciclib, two other highly selective CDK9 inhibitors (see Figure 5b-d and Figure 6). Each of these findings are concordant with our initial results. The relative contribution for different checkpoints in transcriptional control (eg CDK7, CDK9, CDK12/13) and their redundancy vs specificity is an important question and an area of active ongoing research by our team. We feel that dissecting this rigorously, however, is beyond the scope of this present manuscript, which focuses specifically on validating the role of P-TEFb/CDK9.

4. The authors replicated only a part of their findings using zotiraciclib. The addition of Western blotting data (as in Fig.6b) and Caspase3/7 data would strengthen their argument.

We have performed additional radiosensitization, viability, caspase, and Western assays for zotiraciclib. These data are now incorporated into Figure 5e and Extended Data Figure 16.

Minor comments:

1. The correct acronym for Positive transcription elongation factor b is P-TEFb, not PTEFb.

Thank you, we have updated this nomenclature throughout the manuscript.

2. The references in #1 above should be discussed in the introduction section.

We have now included citations and discussions for each of these references. The Barboric and Zhou papers are discussed in the introduction to provide context for P-TEFb/7SK regulation, and the Svejstrup manuscript is discussed in the section corresponding to Figure 2 to provide relevant framing for the observed slowdown of transcriptional rates following DNA damage.

3. Could the authors explain in the text why is SU-DIPG-IV a good cell line model of glioma?

We have added a rationale for the selection of this line. Briefly, it harbors a histone mutation as is a typical driver of pediatric glioma while remaining TP53 WT so as to not confound the mechanistic chromatin studies with TP53-Mt effects. We later repeat functional assays in TP53-Mt models to show that the combinatorial findings hold regardless of TP53 status (Extended Data Figure 9).

4. Could the authors include the concentrations of drugs used in all of their Figure legends? At present, these are mostly non-existent.

We have added drug concentrations to the respective figure legends.

5. For the reader, the legend in Fig.2e could be very useful.

We have added a legend to this figure (now Fig 2f).

6. Fig.3a: is quantification shown normalized to total Pol II levels?

This was initially shown as normalized to tubulin. We have now normalized to total Pol II (and tubulin loading), and we have updated this in the figure and legend.

7. 'Given the role of H3K27ac in recruiting P-TEFb...' – could the authors include the reference to back up this claim?

This transition references the mechanism of BRD4 and SEC recruitment discussed in the prior section and illustrated in Figure 2b. We have added the appropriate references to this statement to clarify this relationship.

8. Fig.3c: the clusters shown need a better explanation.

We have updated the figure legend and results sections regarding these clusters.

9. Fig.3f: a metagene plot would be very helpful here.

We have added a metagene plot to this panel.

10. Fig.4c: BT245 isn't mentioned in the text that refers to this panel.

We have corrected the text to reflect the cell lines shown in the figure.

11. Fig.4d: Could you label the cell line used for these experiments?

We have added the SU-DIPG4 label to the figure and accompanying legend.

12. Fig.5b: Could the authors quantify the synergy (i.e. Bliss score)?

Sensitization enhancement ratio (SER) derived from normalized clonogenic assays is a quantified assessment, and is the preferred in vitro assessment for radiosensitization (as opposed to Bliss/Loewe for drug-drug synergy). See ESTRO statement on "Measurement of the effects of radiosensitizing drugs and evaluation of their statistical significance". We have added the relevant citation to Subiel et al to clarify this.

13. Fig.5c and Extended Fig.6: This needs a better description and presentation of the data, particularly ExtFig.6.

We have significantly reformatted Extended Figure 6 to improve clarity, and we have revised the text and legend accompanying Figure 5c (now 5e). Beyond this, quantifying the ALDH+ fraction and displaying as a percent overlying the flow plot is a standard means of presenting this data, as seen in the accompanying references.

14. ExtFig.7: To appreciate the differences, the images need a better contrast.

We are somewhat limited as these are extracted from live cell imaging recordings. We have now used software to (uniformly) increase sharpness and contrast of the images, but this contributed to our decision to move these to supplemental with appropriate quantifications provided.

15. It would be very interesting to see BETi or SEC-specific compounds in the orthotopic model.

We have addressed the mechanistic question of BRD4 vs SEC specificity or redundancy via additional CUT&RUN experiments (see discussion above) and acknowledged the limitations of the study in this regard in both the Results and Discussion sections. Testing what are largely tool compounds in vivo would test the translational suitability and pitfalls of these specific compounds without necessarily generating new biologic insights for their targets.

16. As IR has been reported to deplete LARP7 via BRCA1-mediated protein degradation that in turn affects cellular DDR (DOI: 10.1016/j.celrep.2020.107974), the authors need to 1) test whether IR depletes LARP7 in their systems, and if so, 2) discuss how this result relates to their key findings.

While we observed a release of CDK9 from LARP7 (Figure 2d), we did not see a degradation of LARP7 protein in our model systems. These data are included in Extended Data Figure 1.

Reviewer #3:

The manuscript entitled "Rapid PTEFb-dependent transcriptional reorganization underpins the glioma adaptive response to radiotherapy" highlights a potential novel therapeutic strategy against diffuse intrinsic pontine gliomas. Overall, the manuscript is executed well. Certain points require improvement:

We appreciate this reviewer's characterization of the manuscript as novel and well executed. Specific points addressed below:

1. Figure 5E requires a cDNA rescue with wt-CDK9 as well as with an inactive mutant.

Despite attempts with multiple construct backbones and transfection methods, these cells do not tolerate dual transfections. As an alternative means of addressing this comment, we expressed either WT-CDK9 or the catalytically inactive D167N mutant prior to treatment with IR +/- AZD4573. Overexpression of WT-CDK9 rescued cells from the radiosensitizing effect of AZD4573, while the D167N mutant did not. This data is now included in Figure 5h.

2. Fig 6B: They should over-express Mcl-1 to demonstrate functional involvement in apoptosis induction in the related context.

We have performed this experiment as suggested and included the data in Extended Data Figure 12. MCL1 overexpression significantly delayed the induction of apoptosis following AZD4573 treatment, and it rescued cell viability from AZD4573 exposure at comparable fixed timepoint. Together these data are consistent with a role for MCL1 in mediating AZD4573's cytotoxic effects.

3. Why do they use a DIPG model in the flank? This does not seem to be an appropriate micro-environment for those types of tumors. Probably, this experiment should be included in the supplement.

We agree with and acknowledge the significant limitations presented by testing brain tumor therapies in subcutaneous models. However, flank models are frequently used in the DIPG literature (PMID: 28968963, PMID: 33106374, PMID: 36274161, PMID: 32832670, PMID: 37603953) to test proof-of-target effects for tool compounds with known blood-brain-barrier limitations. As the majority of the mechanistic work is performed with AZD4573, we feel it is valuable to demonstrate the predicted efficacy in vivo, even acknowledging that this specific agent may not have practical clinical application for this disease. We have included the following language to help clarify this:

“This supports the therapeutic utility of effective on-target CDK9 inhibition, though it highlights a translational limitation of this specific compound in a CNS disease context... While the unique anatomic and physiologic considerations for CNS tumors necessitate the selection of appropriate agents for adequate CNS delivery, our data supports the use of clinically-relevant CDK9 inhibitors in a multimodal treatment approach for these lethal cancers.”

4. Fig 7G: Why are only two images shown?

Candidly, access to our animal facilities was impacted during COVID19-related shutdowns while this study cohort was conducted. As such, we have complete MRI images available for only a few animals. We have added the second available pair imaged at the same timepoints post-injection in Extended Data Figure 17. As these numbers are too few for statistical volumetric comparisons, we have included them only as representative examples of the data available. We believe the images to still be of interest, as the statistical measure of the study is reflected in the survival and BLI data.

REVIEWERS' COMMENTS

Reviewer #1 (Remarks to the Author):

The authors have conducted additional experiments that addressed my previous concerns. I have no additional questions.

Reviewer #2 (Remarks to the Author):

The authors have adequately addressed most of the concerns I raised previously. However, before acceptance, the following issues need to be addressed.

- To address the major comment #1, the authors have IPed CDK9 from cells before and after IR exposure, and probed for the 7SK snRNP component LARP7 (Figure 2d). First, could the authors add the timing and IR conditions to the figure legend? Most importantly, levels of CDK9 in the IPs should be shown. Without this data, the authors cannot conclude that there is a clear decrease in the levels of LARP7 bound to CDK9 in irradiated cells. In addition, as HEXIM1 inhibits P-TEFb within 7SK snRNP, blotting for it would be more appropriate in this experiment.

- To address the major comment #2, the authors have performed immunoblotting for the phosphorylated Ser1627/1631 positions of BRG1 following IR in the presence or absence of AZD4573. From the data, it is not clear which band is the correct one to look at. Thus, the authors would need to clarify this point. In addition, total levels of BRG1 should be shown to reach any conclusions of this experiment.

Reviewer #3 (Remarks to the Author):

The author have addressed parts of the concerns.

The manuscript could still benefit from a quantification of the intracranial tumors, showing that indeed the treatments reduce the sizes of the tumors when compared to the controls.

REVIEWER #1

The authors have conducted additional experiments that addressed my previous concerns. I have no additional questions.

We appreciate the prior suggestions and are glad that we were able to satisfactorily address the reviewer concerns.

REVIEWER #2

The authors have adequately addressed most of the concerns I raised previously. However, before acceptance, the following issues need to be addressed.

- To address the major comment #1, the authors have IPed CDK9 from cells before and after IR exposure, and probed for the 7SK snRNP component LARP7 (Figure 2d). First, could the authors add the timing and IR conditions to the figure legend? Most importantly, levels of CDK9 in the IPs should be shown. Without this data, the authors cannot conclude that there is a clear decrease in the levels of LARP7 bound to CDK9 in irradiated cells. In addition, as HEXIM1 inhibits P-TEFb within 7SK snRNP, blotting for it would be more appropriate in this experiment.

We have optimized and repeated the co-IP experiment, including blots for CDK9, HEXIM1, and MEPCE now in addition to LARP7. This data, along with the requested legend information, is updated in Figure 2d.

- To address the major comment #2, the authors have performed immunoblotting for the phosphorylated Ser1627/1631 positions of BRG1 following IR in the presence or absence of AZD4573. From the data, it is not clear which band is the correct one to look at. Thus, the authors would need to clarify this point. In addition, total levels of BRG1 should be shown to reach any conclusions of this experiment.

We have confirmed the appropriate molecular weight for p-BRG1 and cropped the image for clarity. We have additionally probed for total BRG1 and added this, as well as quantifications for the ratio of p-BRG1 to total BRG1. This image should now be more readable while better conveying a CDK9-dependent increase in BRG1 phosphorylation following IR exposure.

REVIEWER #3

The author have addressed parts of the concerns.

The manuscript could still benefit from a quantification of the intracranial tumors, showing that indeed the treatments reduce the sizes of the tumors when compared to the controls.

To fully address this, we have injected and treated additional animals and performed MRIs at the same time points as those previously shown in sufficient numbers to allow statistical comparison. Both the original and new MRIs have undergone volumetric analysis by a radiologist blinded to the treatment assignment of the animals. These new images and quantifications are now included in Figure 7g and Extended Data Figure 17.